# Murine hematopoietic progenitor cell lines with erythroid and megakaryocyte potential

Ruiqiong Wu[1,9], Faraz Salehi [2,9], Vanessa Redecke[2], Zhijun Ma[3], Marco Marchetti[4], David Finkelstein[5], Peng Xu[1,8], Yong Cheng[1], Kimberly A. Queisser[6], Aaron C. Petrey[6], Conroy O. Field[7], Hyun Sook Ahn[7], Mortimer Poncz [7], Mitchell J. Weiss [1] & Hans Häcker [2] ✉

Red blood cells and platelets derive from bi-potential bone marrow megakaryocyte-erythroid progenitors, but their study is constrained by cell scarcity and limited experimental systems. Here we show that conditional expression of a virally transduced, regulated form of Hoxa7 enables expansion of murine cells resembling megakaryocyte-erythroid progenitors (Hoxa7-TPO), which undergo erythro-megakaryocytic differentiation upon *Hoxa7* inactivation. The close relationship of Hoxa7-TPO cells to megakaryocyte-erythroid progenitors is supported by genetic and phenotypic analyses, and mature Hoxa7-TPO-derived red blood cells and platelets are largely indistinguishable from their primary counterparts. Genetic knock-out studies in Hoxa7-TPO cells recapitulate the key function of *Klf1* and *Nfe2* in red blood cell and platelet development, respectively, while disruption of the von Willebrand receptor gene *Gp1ba* recapitulates features of human Bernard-Soulier syndrome. Hence, we developed a versatile experimental system for expansion and differentiation of megakaryocyte-erythroid progenitors to study red blood cell and platelet development and model human diseases.

Adult hematopoiesis is initiated in the bone marrow (BM) from hematopoietic stem cells (HSC). HSCs restrict their lineage potential progressively, producing intermediate multipotent progenitors (MPPs) that eventually differentiate via intermediate cell types into mature blood cells. One well-characterized intermediate cell type is the Megakaryocyte (MK)-Erythroid progenitor (MEP), which gives rise to red blood cells (RBC) and MK/ platelets, but lacks lineage potential for major immune cell types, such as macrophages, granulocytes and lymphocytes[1,2]. Given the key effector functions of RBC and platelets, i.e., oxygen transport and primary hemostasis, respectively, regulation

of MEP biology is critical for balanced hematopoiesis and human health. Key factors that control cell differentiation of MEP towards RBC and platelets have been defined. However, more detailed investigations are difficult due to a lack of suitable experimental systems. We previously showed that a combination of regulated HOXB8 and defined growth factors conditionally immortalize macrophage- and neutrophil-committed progenitors or multipotent progenitors with myeloid and lymphoid potential[3,4]. This approach takes advantage of the well-characterized function of Hox genes to block cell differentiation in favor of self-renewal, which is also the basis for the

[1]Department of Hematology, St. Jude Children's Research Hospital, Memphis, TN, USA. [2]Laboratory of Innate Immunity and Signal Transduction, Department of Pathology, Division of Microbiology & Immunology, University of Utah School of Medicine, Salt Lake City, UT, USA. [3]Department of Bone Marrow Transplantation, St. Jude Children's Research Hospital, Memphis, TN, USA. [4]Department of Human Genetics, Utah Center for Genetic Discovery, University of Utah, Salt Lake City, UT, USA. [5]Department of Computational Biology, St. Jude Children's Research Hospital, Memphis, TN, USA. [6]University of Utah Molecular Medicine Program, Department of Pathology, Division of Microbiology & Immunology, University of Utah School of Medicine, Salt Lake City, Utah, USA. [7]Department of Pediatrics, Division of Hematology, Children's Hospital of Philadelphia, Philadelphia, PA, USA. [8]Present address: Cyrus Tang Medical Institute, Soochow University, Suzhou, The People's Republic of China. [9]These authors contributed equally: Ruiqiong Wu, Faraz Salehi. ✉e-mail: hans.haecker@path.utah.edu

established oncogenic role of various Hox family members[5]. This property is also well illustrated by *Hoxa7*. Genetic deficiency of *Hoxa7* in mice results in reduced numbers of MEPs, while inappropriate expression is a feature of leukemias harboring mixed-lineage leukemia (MLL) mutations, reflecting reduced and increased self-renewal, respectively[6,7].

Here we show that a combination of an estrogen-regulated form of Hoxa7 and thrombopoietin (TPO) can be used to conditionally immortalize MEP-like cells, which can be expanded in vitro for extended periods of time in the presence of estrogen (activating Hoxa7) and differentiate faithfully into RBC and platelets upon estrogen withdrawal (inactivating Hoxa7) in the presence of suitable growth factors in vitro or in vivo.

## Results
### Generation of Hoxa7-TPO cells
To investigate if MEP could be targeted by a regulated Hox gene, we transduced largely unfractionated BM cells, which had briefly been expanded in vitro, with a murine stem cell virus (MSCV)-based retroviral vector expressing an estrogen-activated form of *Hoxa7*. The fusion construct, ERHBD-Hoxa7, consists of *Hoxa7* fused to the hormone-binding domain of the estrogen receptor (ERHBD) containing the characterized G400V substitution, which renders the hormone-binding domain insensitive to physiological concentrations of estrogen or phenol red contained in growth medium[8]. As noted, *Hoxa7* was previously shown to cause selective deficiency of MEPs and, therefore, we hypothesized that its sustained expression might support expansion of MEPs[6]. As expected, cells transduced with a control vector eventually stopped growing in the presence of estrogen and TPO, whereas treatment of ERHBD-Hoxa7-transduced cells with estrogen (activating *Hoxa7*) and TPO led to exponential expansion of

blast-like cells, whose survival was strictly dependent on TPO (Fig. 1a–d). These cells proliferated for extended periods of time, and karyotype analysis after one year of continuous cell culture showed no detectable abnormalities (Supplementary Fig. 1a, b). Thus, TPO in context with conditionally activated Hoxa7 can be used to generate genetically stable, growth-factor-dependent cell lines.

### Phenotype analysis and megakaryocyte potential of Hoxa7-TPO cells in vitro
Consistent with their blast-like morphology, flow cytometry analysis showed that Hoxa7-TPO cells expressed the progenitor marker c-kit at high levels, but were negative for cell lineage markers, including Ter119, CD41, CD16/32, B220, CD11c, CD115, CD4, and CD8 (Fig. 1e and Supplementary Fig. 2). CD34 and Sca-1, which are expressed on early hematopoietic stem and progenitor cells (HSPC) were not expressed at detectable levels. Interestingly, CD150, which is expressed at high levels in progenitor cells with MK and erythroid potential (Pre MegE) and CD105, which is increasingly up-regulated during RBC development, were expressed at robust levels on Hoxa7-TPO cells (Fig. 1e)[9].

To determine if Hoxa7-TPO cells could differentiate in vitro into MKs, we removed estrogen (to deactivate Hoxa7) and continued to culture the cells in medium containing TPO. Under these conditions the cells ceased to proliferate, and after 4 days, they had formed large cells with up to 16 nuclei (32 N) that were indistinguishable from primary MKs (Supplementary Fig. 3b and Fig. 2a–c). The homogeneity of Hoxa7-TPO-derived cells allowed direct analysis without further enrichment, while BM-derived MKs required enrichment by BSA gradient, a commonly used method to increase homogeneity of primary MKs[10]. Similar to BM-derived MKs, Hoxa7-TPO-derived MKs produced proplatelets, which could be detected by phase contrast microscopy, but were particularly apparent when analyzed by differential

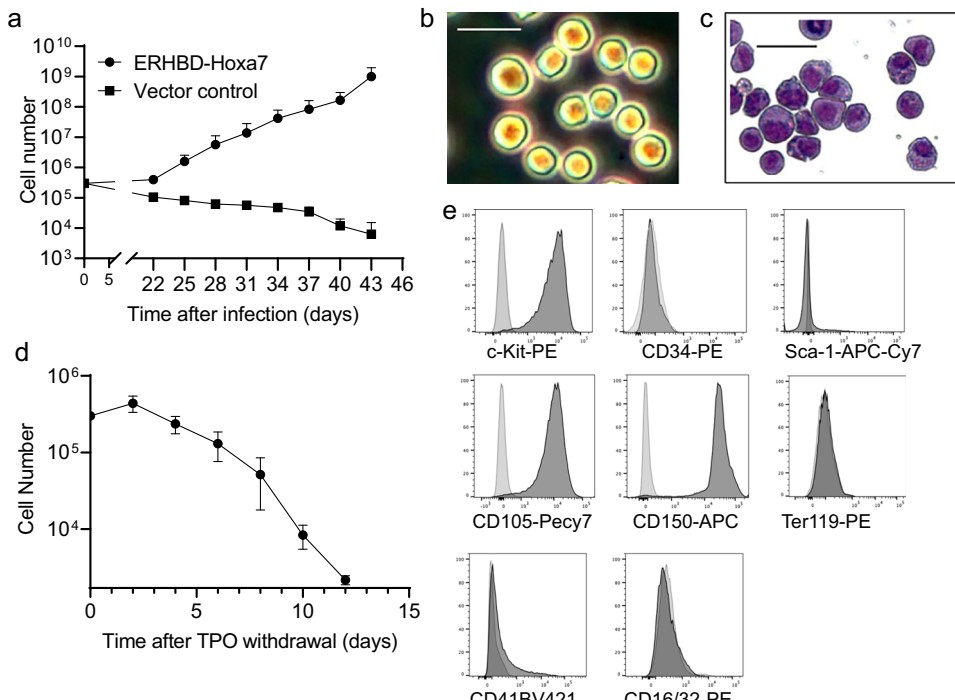

**Fig. 1 | Generation of conditionally immortalized hematopoietic progenitor cell lines based on *Hoxa7* and TPO. a** Growth characteristics of Hoxa7-TPO cells. BM cells from H2K GFP transgenic mice were transduced with an estrogen-regulated form of *Hoxa7* (ERHBD-Hoxa7) or a vector control and cultured in the presence of estrogen and TPO. Numbers of live cells were determined at the time points indicated. Source data are provided as a Source Data file. **b** Phase contrast microscopy of stably growing Hoxa7-TPO cells. **c** Cytology analysis of Hoxa7-TPO

cells based on May-Grünwald-Giemsa staining. **d** Cell survival analysis of Hoxa7 cells in the absence of TPO. TPO was withdrawn at day 0 and the number of live cells was determined during time. Source data are provided as a Source Data file. Scale bars (B, C, E) = 20 µm. Error bars represent standard deviation of five biological replicates. **e** Flow cytometry analysis of Hoxa7-TPO cells. Dark grey, indicated antibodies; light grey, isotype controls.

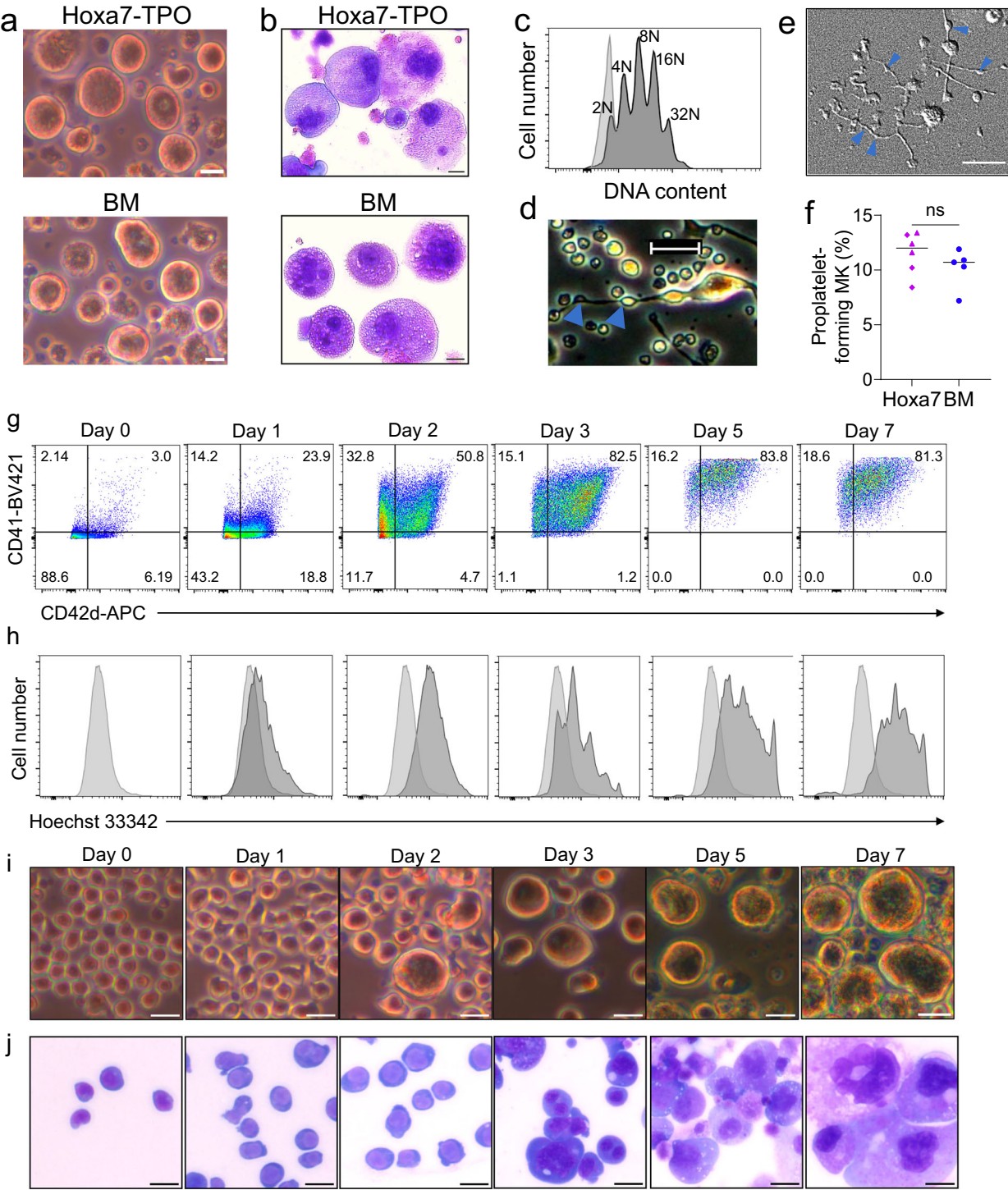

**Fig. 2 | Phenotype analysis and in vitro MK differentiation of Hoxa7-TPO cells.**
**a** Phase contrast images of in vitro differentiated Hoxa7-TPO and BM cells cultured in the presence of TPO in the absence of exogenous estrogen for 4 and 6 days, respectively. **b** May-Grünwald-Giemsa staining of Hoxa7-TPO and BM cells described in (**a**). **c** Flow cytometry analysis of Hoxa7-TPO cells that were left non-differentiated (light grey) or were differentiated for 4 days in the presence of TPO using the DNA dye Hoechst H33342. **d** Phase contrast image of in vitro differentiated Hoxa7-TPO cells cultured in the presence of TPO in the absence of exogenous estrogen for 7 days. Proplatelets are indicated by blue arrows.
**e** Differential interphase contrast (DIC) image of in vitro differentiated Hoxa7-TPO cells in the presence of TPO highlighting proplatelet formation (blue arrows).
**f** Quantification of proplatelet-producing MK described in e in comparison to BM-

derived MK, which were differentiated in the presence of TPO for 4 days and enriched by BSA-gradient. Data represent $n = 6$ independent experiments for Hoxa7-derived MK and $n = 5$ for BM-derived MK. Purple triangles and diamonds indicate independently established Hoxa7-TPO populations. Ns, not significant; Two-tailed unpaired t test. Source data are provided as a Source Data file. **g**, **h** Flow cytometry analysis of in vitro differentiating Hoxa7-TPO cells in the presence of TPO using antibodies against CD41 and CD42d (**g**) and the DNA dye Hoechst 33342 (**h**). Light gray, non-differentiated cells; dark grey, differentiating cells. **i**, **j** Phase contrast (**i**)- and May-Grünwald-Giemsa (**j**)-based microscopy images of in vitro differentiating Hoxa7-TPO cells in the presence of TPO. Magnification 20x. Scale bars (**a**, **b**, **d**, **e**, **i**, **j**) = 20 μm.

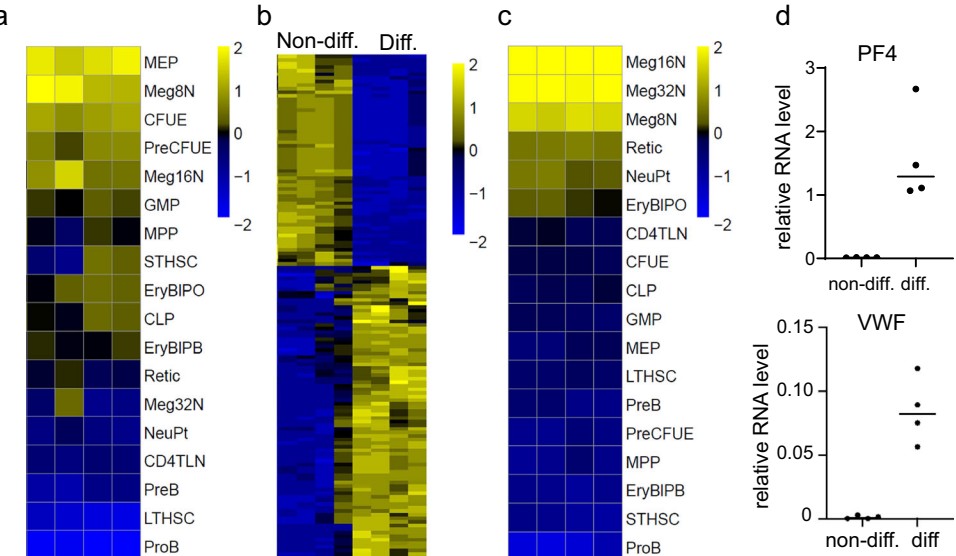

**Fig. 3 | Transcriptomics-based comparative analysis of Hoxa7-TPO cells and primary hematopoietic cell types. a** Heatmap based on z-scores of averaged log 2 FPKM gene expression for each signature that were derived from four populations each of non-differentiated Hoxa7-TPO cells and defined, murine hematopoietic cell types obtained from Haemopedia[13]. **b** Heatmap displaying the z-score transformation of average of log 2 FPKM of highly expressed genes for each cell type. The genes were selected by RNAseq analysis derived from four populations of non-differentiated Hoxa7-TPO cells and Hoxa7-TPO cells that were differentiated in TPO for four days. **c** Heatmap based on z-scores of averaged log 2 FPKM gene expression for each signature derived from four populations of Hoxa7-TPO cells that were differentiated with TPO for four days and defined, murine hematopoietic cell types

obtained from Haemopedia[13]. **d** Q-PCR analysis of four populations of non-differentiated Hoxa7-TPO cells and Hoxa7-TPO cells that were differentiated in TPO for four days. Source data are provided as a Source Data file. LTHSC, Long-term hematopoietic stem cells; STHSC, short-term hematopoietic stem cells; MPP, multipotent progenitors; GMP, granulocyte macrophage progenitors; CLP, common lymphoid progenitors; PreCFUE, Pre-colony forming unit – erythroid; MEP, megakaryocyte erythrocyte progenitors; CFUE, colony forming unit – erythroid; EryBlPB, polychromatic (some basophilic) erythroblasts; EryBlPO, polychromatic (some orthochromatic) erythroblasts; Retic, reticulocytes; Meg8N, 8N MK; Meg16N, 16N MK; Meg32N, 32N MK; NeutPt, neutrophils from peritoneal cavity; ProB, Pro-B cells; PreB, Pre-B cells; CD4TLN: CD4 T cell from lymph nodes.

interphase contrast (DIC) microscopy (Fig. 2d, e)[11,12]. Quantitative analysis showed a similar percentage of Hoxa7-TPO- and BM-derived MKs participating in proplatelet formation (Fig. 2f). A more detailed kinetics analysis of Hoxa7-TPO cells during TPO-driven cell differentiation using flow cytometry (CD41, CD42d, Hoechst H33342) and microscopy showed that the cells start to increase their size already by day 1 after estrogen withdrawal, with increasing surface expression of CD41 and CD42d at day 2 (Fig. 2g–j and Supplementary Fig. 3b). Cells containing more than one nucleus indicative of endomitosis became apparent at day 3, after which cell differentiation was rapidly progressing and largely completed by day 5, although polyploidy was further increased at day 7 (Fig. 2h). As noted, a particularly obvious aspect of this process employing Hoxa7-TPO cells is the homogeneity of the mature MK cell population generated, which sets it apart from primary BM cultures. Together, the data suggest that Hoxa7-TPO cells have MK potential that can be realized in vitro to generate homogenous, mature MK cell populations of essentially unlimited cell numbers that resemble phenotypically primary MKs.

## Transcriptional analysis of Hoxa7-TPO cells and primary hematopoietic cell types

To further characterize Hoxa7-TPO cells, we performed RNAseq analyses of four independently established bulk cell populations, either in their non-differentiated state or after differentiation in the presence of TPO for four days. Interestingly, gene expression-based correlation analyses with defined primary hematopoietic cell populations (available through Hemopedia[13]) showed the strongest correlation of non-differentiated Hoxa7-TPO cells with bi-potent megakaryocyte/ erythroid progenitors (MEP), followed by immature MK and colony-forming units-erythroid (CFUE). Less differentiated HSPC, including long-term hematopoietic stem cells (LT-HSC) and MPP, as well as myeloid progenitors (granulocyte monocyte

progenitors (GMP) and lymphoid progenitors (common lymphoid progenitors (CLP), Pro B-cells) were more distantly related (Fig. 3a). Global comparison of non-differentiated vs. differentiated cells illustrated a massive change in gene expression (Fig. 3b and Supplementary Data 1). Cells differentiated in the presence of TPO had lost apparent similarity to progenitor cells, but exhibited a very strong correlation with MKs, followed by reticulocytes (Fig. 3c). Up-regulation of two MK signature genes, *Pf4* and *Vwf*, in TPO-differentiated Hoxa7-TPO cells was confirmed by Q-PCR (Fig. 3d). The fundamental changes in gene regulation during cell differentiation were recapitulated on the level of lineage-defining factors, including *Mpl* and *Itga2b*, as expected (Supplementary Fig. 4). They were also recapitulated by gene ontology analyses, illustrating for example up-regulation of autophagy-related genes and down-regulation of RNA synthesis-related genes in mature MK (Supplementary Fig. 5 and Supplementary Data 2). This analysis also shows the down-regulation of mitotic cell cycle-related genes in mature MK (and their up-regulation in RBC, see below), which is consistent with the negative correlation of cell cycle kinetics and MK differentiation of primary MEP (Supplementary Fig. 5)[14]. Taken together, the data show that Hoxa7-TPO-derived mature MKs resemble transcriptionally their primary counterparts. The close relationship with MEP and co-expression of CD105 and CD150 indicated that Hoxa7-TPO cells might have both MK and erythroid potential.

## In vivo potential of Hoxa7-TPO cells

To evaluate the in vivo lineage potential of Hoxa7-TPO cells, we established cell lines using BM from GFP-transgenic mice, which express GFP in all hematopoietic lineages, facilitating their identification by flow cytometry[15]. Cells grown in the presence of TPO and estrogen were adoptively transferred to lethally irradiated mice (along with a small number of GFP-negative BM helper cells from isogenic

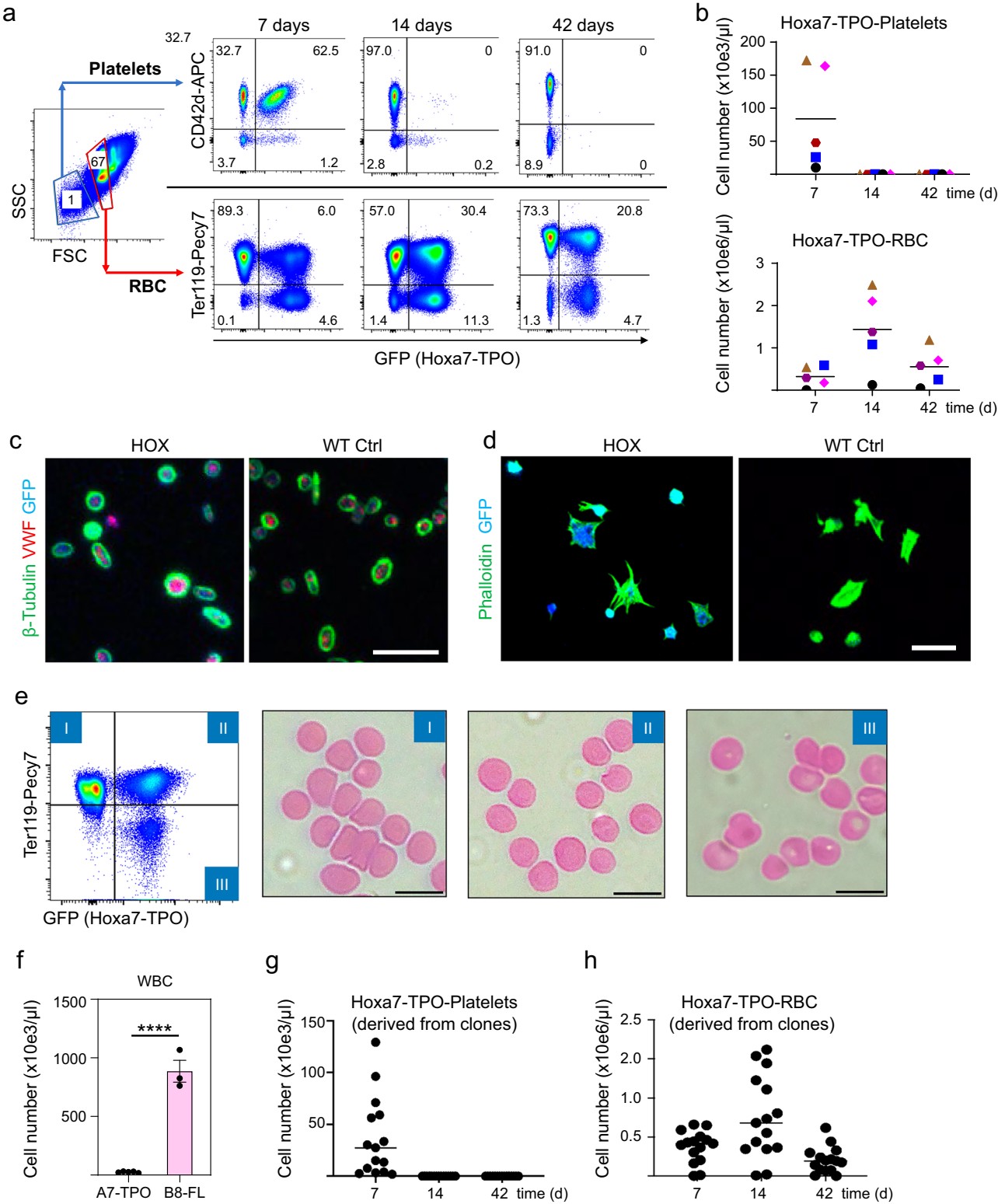

mice), and the generation of GFP⁺ Hoxa7-TPO-derived cells was analyzed in the peripheral blood (PB) over time. One week after transfer, substantial levels of Hoxa7-TPO-derived platelets were apparent in all mice adoptively transferred with independently established Hoxa7-TPO cell populations (Fig. 4a, b). Cell size of Hoxa7-TPO-derived platelets was comparable to BM-derived platelets under various conditions, including mice transferred with Hoxa7-TPO cells and mice transplanted with GFP + BM to mimic adoptive transfer conditions (Supplementary Fig. 6). Owing to the low number of BM cells

transferred (2x10e5 cells) and the early time point of analysis, endogenous, host-derived platelets in Hoxa7-TPO-transferred mice likely represent exclusively cells that survived irradiation, rather than de-novo, donor-derived cells. To analyze morphological and functional aspects of Hoxa7-TPO-derived platelets, we transferred non-differentiated Hoxa7-TPO cells into PF4-Cre-iDTR mice, in which MK and platelets can be eliminated efficiently by in vivo administration of diphtheria toxin (DT), thereby increasing the percentage of Hoxa7-TPO-derived mature platelets (see Methods section). 6 days after

**Fig. 4 | In vivo potential of Hoxa7-TPO cells. a** Flow cytometry analysis of the PB of mice that were adoptively transferred with GFP-expressing Hoxa7-TPO cells for indicated time periods. **b** Quantitative analysis of Hoxa7-TPO-derived platelets and RBC in the PB of mice adoptively transferred with independently established Hoxa7-TPO cell populations (*n* = 6) for indicated time periods. Symbols represent independently established cell populations. Source data are provided as a Source Data file. **c, d** Immuno-fluorescence analysis of washed Hoxa7-TPO-derived and endogenous platelets that were seeded on chamber slides coated with Poly-L-lysine (**c**) or fibrinogen (**d**), incubated with antibodies against indicated proteins or phalloidin, followed by confocal microscopy. Scale bars = 10 μm. **e** Flow cytometry-based gating strategy (left) and microscopy images (right) of May-Grünwald-stained cytospins of FACS-purified cell populations 14 days after adoptive transfer of Hoxa7-TPO cells. I = BM-derived RBC; II = Hoxa7-TPO-derived Ter119[+] RBC; III = Hoxa7-TPO-derived Ter119[-] RBC. Scale bars = 10 μm. **f** Flow cytometry-based quantification of Hoxa7-TPO- and Hoxb8-FL-derived white blood cells (WBC) in the PB 7 days after adoptive transfer. Data represent mean ± SD for 5 (Hoxa7-TPO) and 3 (Hoxb8-FL) adoptively transferred mice. ****P < 0.0001. Two-tailed unpaired t test. Source data are provided as a Source Data file. **g, h** Quantitative analysis of platelets (**g**) and RBC (**h**) in the PB of mice that were adoptively transferred with 15 clones derived from Hoxa7-TPO cells. Each symbol represents data obtained from individual mice transferred with one individual clone. Source data are provided as a Source Data file.

transfer, platelets were isolated and analyzed by immuno-fluorescence with antibodies against β-tubulin, von Willebrand factor (vWF) and GFP, the latter identifying Hoxa7-TPO-derived cells. As shown in Fig. 4c, Hoxa7-TPO-derived cells showed a typical platelet morphology as reflected by a classic tubulin ring surrounding granular vWF, similar to endogenous platelets of untreated control mice. Moreover, when plated on fibrinogen, both Hoxa7-TPO-derived and endogenous platelets exhibited characteristic, actin-dependent spreading of platelets (Fig. 4d).

Hoxa7-TPO-derived platelets largely disappeared by day 14 after transfer, consistent with terminal differentiation accompanied by loss of self-renewal of Hoxa7-TPO cells in the absence of exogenous estrogen and the relatively short half-life of mouse platelets (Fig. 4a)[16]. In contrast, GFP+ Ter119+ RBC were detected at day 7 after Hoxa7-TPO cell transfer, reached peak levels at day 14 and remained present after six weeks, reflecting the 38 to 52 day in vivo half-life of RBC[17] (Fig. 4a, b). Morphologically Hoxa7-TPO-derived RBC were indistinguishable from endogenous cells (Fig. 4e). We noted a population of GFP+ Ter119- RBC, which was detected primarily at early time points and progressively lost thereafter (Fig. 4a). Morphologically, these cells were very similar to Ter119+ RBC (Fig. 4e). We speculate that these cells represent less mature RBC, which may be generated under the very particular conditions in irradiated mice. This interpretation is consistent with the smaller percentage of these cells found in endogenous RBC populations, which consist largely of aged cells that persist after lethal irradiation. The Hoxa7-TPO cells produced very few white blood cells (WBC) in vivo, in contrast to Hoxb8-FL cells, which represent conditionally immortalized WBC progenitors, but lack potential for platelet and RBC development[4] (Fig. 4f). To exclude the possibility that the bi-lineage potential of Hoxa7-TPO cell populations was due to a mixture of cells with lineage potential for either platelets or RBC, we established single cell clones and evaluated their lineage potential in adoptive transfer experiments. Most clones maintained both platelet and RBC potential, similar to what we observed in bulk cell (non-clonal) populations (Fig. 4g, h). While the overall lineage output of individual clones varied, platelet and RBC potential correlated quite well, as reflected by a Pearson correlation index of 0.46 (Supplementary Fig. 7). As such, some clones appeared to possess an overall higher lineage potential. However, we note that the cloning procedure itself may exert stress on the cell lines, which may contribute to differences in linage output. Cryopreserving established Hoxa7-TPO cells in liquid nitrogen did not reduce cell performance as assessed by recovery and adoptive transfer of Hoxa7-TPO cells that had been cryopreserved for 4 years in liquid nitrogen (Supplementary Fig. 8a–c). Together with results obtained by flow cytometry and gene expression analyses, the data suggest that Hoxa7-TPO cells correspond closely to primary MEP, with potential for megakaryocyte and erythroid lineages.

### Functional analysis of Hoxa7-derived platelets
Thrombin treatment of platelets induces their activation markers and aggregation, reflecting essential, physiological functions. To determine if Hoxa7-TPO-derived platelets maintained these effector functions, we analyzed platelets from the PB of adoptively transferred mice in comparison to endogenous platelets from the same mice. Hoxa7-TPO-derived platelets showed thrombin-induced CD62P up-regulation comparable to endogenous platelets (Fig. 5a and Supplementary Fig. 8d). Moreover, thrombin treatment resulted in aggregation of Hoxa7-TPO-derived platelets, reflected by loss of single platelets, which was largely comparable to their endogenous counterparts (Fig. 5b). Slight differences may be due to the fact that Hoxa7-TPO-derived platelets represent newly generated (younger) cells, while endogenous platelets are largely composed of older cells that persist after lethal irradiation. Thus, the data suggest that Hoxa7-derived platelets are functionally active.

To confirm and extend the functional activity of the Hoxa7-TPO-derived platelets, we studied their hemostatic efficacy in a cremaster arteriole vascular injury in situ microscopy model in which thrombus formation can be visualized in real time[18]. To limit the contribution of the endogenous platelets, we used previously described *Mpl*[-/-] mice as the recipients, in which platelet counts are reduced by approximately 60–80% compared to wildtype (WT) C57B/6 mice[19]. Three distinct groups of mice were studied, i.e., (i) untreated *Mpl*[-/-] control mice, (ii) irradiated *Mpl*[-/-] control mice receiving only a small number of wildtype helper BM cells, and (iii) irradiated *Mpl*[-/-] mice receiving wildtype BM helper cells plus non-differentiated Hoxa7-TPO cells (Fig. 5c). Whole blood flow cytometry analysis from these mice showed that the relative number of platelets was ~4–6% of the total cells. In mice transplanted with BM plus Hoxa7-TPO cells, roughly two-thirds of platelets were derived from the Hoxa7-TPO cells (Fig. 5d). All three groups were studied for hemostatic efficacy by in situ microscopy of clot formation following laser injury to cremaster arterioles. Compared to untreated *Mpl*[-/-] mice or *Mpl*[-/-] mice transplanted with only BM helper cells, the concurrent presence of BM- and Hoxa7-TPO-derived platelets led to a more vigorous thrombus formation at the site of injury, with Hoxa7-TPO-derived platelets contributing the majority of clot formation (~60%)(Fig. 5c–g and Supplementary Fig. 9). Interestingly, the presence of Hoxa7-TPO-derived platelets increased the incorporation of endogenous (*Mpl*-deficient) platelets into thrombi significantly, likely compensating for the functional deficiency of *Mpl*-deficient platelets (Fig. 5f, g). Thus, Hoxa7-TPO-derived platelets exhibit hemostatic function, as evidenced by their contribution to thrombus formation after arteriole vascular injury.

### RBC development of Hoxa7-TPO cells in vivo
We isolated Hoxa7-TPO-derived reticulocytes from mice and analyzed their gene expression profile, which confirmed their close relation to primary reticulocytes and dissimilarity to other hematopoietic cell types (Fig. 6a)[20]. Similar to Hoxa7-TPO-derived MK, the gene expression profile of Hoxa7-TPO-derived reticulocytes was fundamentally different from the parent undifferentiated cells grown in estrogen-containing medium. In particular, differentiation to the RBC lineage was associated with induction of signature erythroid genes, such as *Zfpm1, Alad, Hbb* and *Hba* and regulation of proteolysis- and RNA-metabolic processes identified by Gene Ontology analysis (Supplementary Figs. 4 and 5). As noted, mitotic cell cycle-related genes were up-regulated in RBC compared to undifferentiated cells, consistent

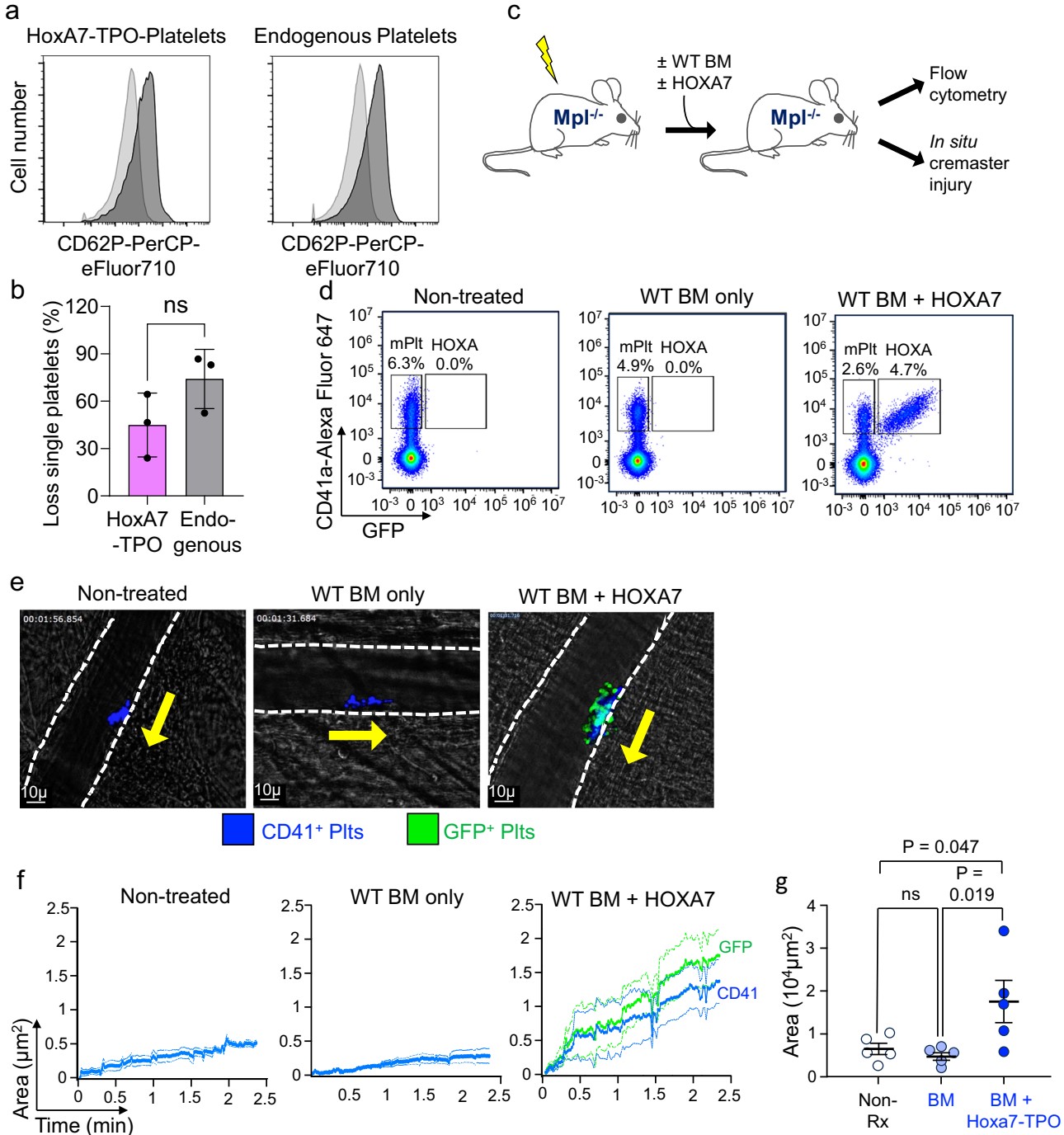

**Fig. 5 | Functional analysis of Hoxa7-TPO-derived platelets. a** Flow cytometry of thrombin-treated platelets. Platelet-rich plasma was prepared from mice six days after adoptive transfer with GFP⁺ Hoxa7-TPO cells and BM-derived helper cells, followed by thrombin treatment (2U/ml) for 20 min. Light grey, non-treated cells; dark gray, thrombin-treated cells. **b** Collagen-induced aggregation analysis of Hoxa7-TPO-derived and endogenous platelets. Whole blood obtained from mice six days after adoptive transfer with GFP⁺ Hoxa7-TPO cells and BM-derived helper cells was treated with collagen (2 µg/ml) for 5 min. The loss of single platelets (reflecting aggregation) was determined by flow cytometry in comparison to non-treated samples. Error bars represent mean ± SD for three replicates obtained from independently transferred mice; Two-tailed unpaired t test. Source data are provided as a Source Data file. **c** Experimental layout of the cremaster arteriole injury model. *Mpl*⁻/⁻ mice were left untreated (non-treated) or irradiated and then infused with either wildtype (WT) bone marrow cells (BM) alone or BM cells along with non-differentiated Hoxa7-TPO cells. **d** Representative flow cytometry analysis of 4 studies indicating endogenous platelets (mPlt) and Hoxa7-TPO-derived platelets

(HOXA) 5 days after adoptive transfer. See Supplementary Fig. 9 for detailed numbers of cells in the various mouse groups. **e** Representative wide-field images at 1.5-2 minutes post injury from videos of laser-induced cremaster arteriole thrombi with CD41⁺ cells in blue and GFP⁺ cells in green. The vessel wall is indicated by the dashed white lines. Direction of flow is indicated by the yellow arrow. Scale bar, 10 µm. **f** Mean ±1 SEM of area-under-the-curve (AUC) of CD41⁺ and CD41⁺GFP⁺ platelet incorporation into thrombi over 2.5 minutes post-laser injury in mice transferred with Hoxa7-TPO cells and BM cells. *N* = 5 with 4-8 injuries per arm is shown. Source data are provided as a Source Data file. **g** Thrombi size (mean ± 1 SEM) for *n* = 5 studies per arm quantified based on total AUC of CD41⁺ platelets over the 2.5-minute study shown in (f) for non-treated *Mpl*⁻/⁻ mice (Non-Rx), mice receiving only BM (BM) and mice receiving BM plus Hoxa7-TPO cells (BM + Hoxa7-TPO). The total CD41 signal includes CD41⁺ BM- and Hoxa7-TPO-derived platelets. Statistical difference was calculated by one-way ANOVA. Ns, not significant. Source data are provided as a Source Data file.

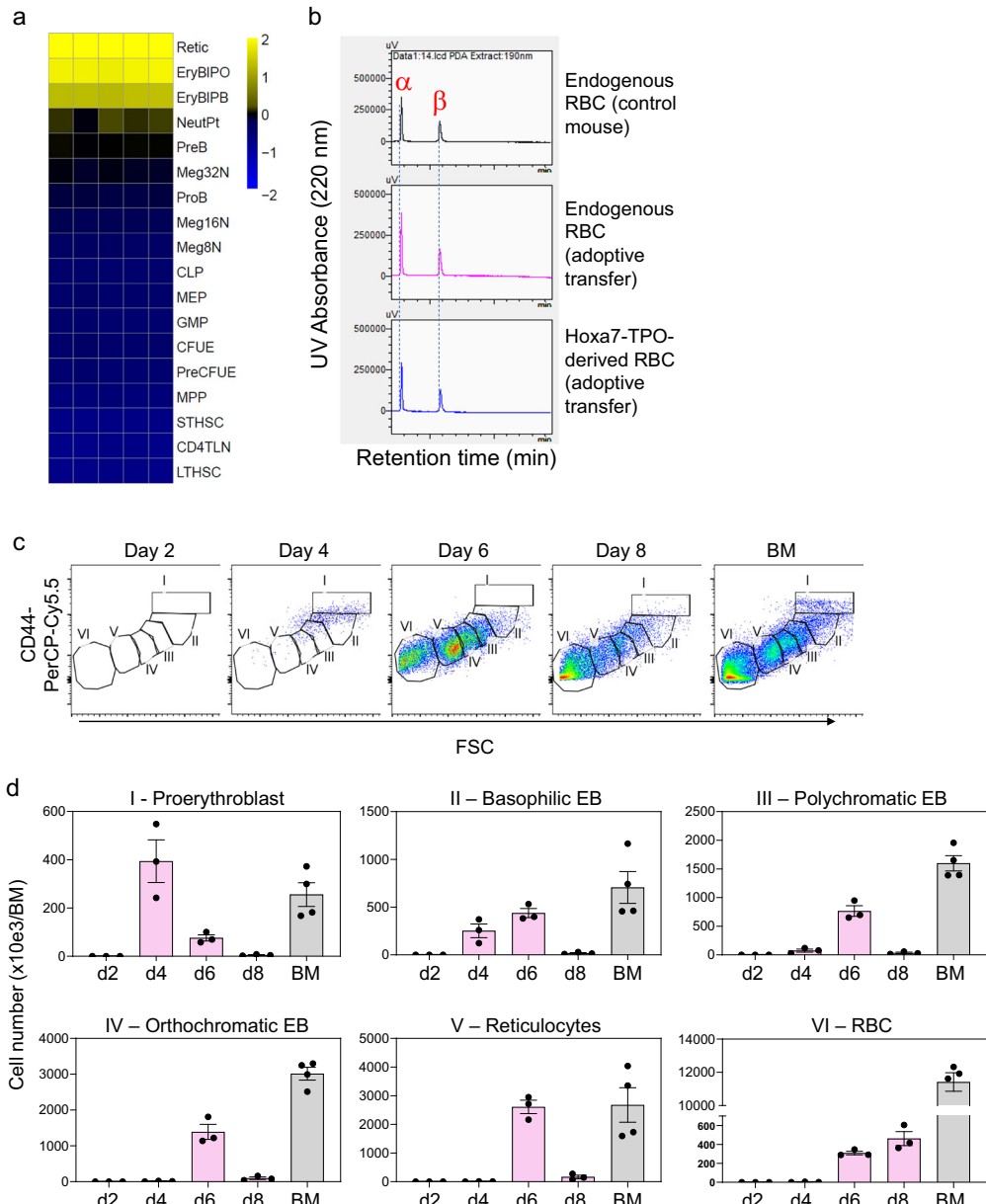

**Fig. 6 | Analysis of Hoxa7-TPO-derived RBC. a** Heatmap based on RNAseq-derived mRNA gene signatures and Pearson's correlation from four populations of Hoxa7-TPO-derived reticulocytes and defined murine hematopoietic cell types obtained from Haemopedia[13]. **b** Chromatography-based hemoglobin analysis of RBC isolated by FACS. Upper panel: RBC from control mice. Middle and lower panel: RBC from adoptively transferred mice, either Hoxa7-TPO-derived cells (GFP⁺, lower panel) or endogenous RBC (middle panel). **c** Flow cytometry analysis of GFP⁺ Hoxa7-TPO cells during terminal erythropoiesis in the BM of adoptively transferred mice at indicated time points. Lin⁻ Ter119⁺ cells are shown. BM from untreated control mice is shown for comparison. I, proerythroblast; II, basophilic erythroblast (EB); III, polychromatic EB; IV, orthochromatic EB; V, reticulocytes; VI, RBC. **d** Quantification of data shown in (**c**). Cell numbers are based on cells isolated from femurs and tibias of mice. Data represent mean and stdev of three mice per time point. Source data are provided as a Source Data file.

with the positive correlation of cell cycle kinetics and RBC differentiation (Supplementary Fig. 5)[14]. Hoxa7-TPO-derived RBC exhibited a classic hemoglobin α/β profile characteristic of adult-type mouse RBC, similar to endogenous RBC (Fig. 6b). Last, based on established maturation steps defined using forward scatter and CD44 expression of Ter119-positive cells, we analyzed the maturation of GFP + , Hoxa7-TPO-derived erythroid progenitors over time[21]. While Hoxa7-TPO cells were merely detectable in the BM at day 1-3 after adoptive transfer, proerythroblasts appeared at day 4, then declined in frequency two days later (Fig. 6c, d). Basophilic RBC appeared on day 4 and day 6, followed by the appearance of polychromatic and orthochromatic erythroblasts. Reticulocytes and mature RBC were detected on day 6 after adoptive transfer (Fig. 6c, d). These data show that after adoptive

transfer, Hoxa7-TPO cells can form committed erythroid progenitors that undergo synchronous terminal maturation with a trajectory that defines normal erythropoiesis. Moreover, the resultant RBCs exhibit normal morphology and hemoglobin content and in vivo lifespan.

## RBC development of Hoxa7-TPO cells in vitro

Given the RBC potential of Hoxa7-TPO cells in vivo, we tested if Hoxa7-TPO cells could also differentiate into RBC in vitro simply by removing estrogen and culturing the cells in the presence of erythropoietin (EPO). Indeed, as assessed by flow cytometry, the cells survived largely in the presence of EPO and started to reduce CD44 expression after day 4, accompanied by up-regulation of Ter119 in ~25% of the cells (Fig. 7a and Supplementary Fig. 3c). This phenotype was also

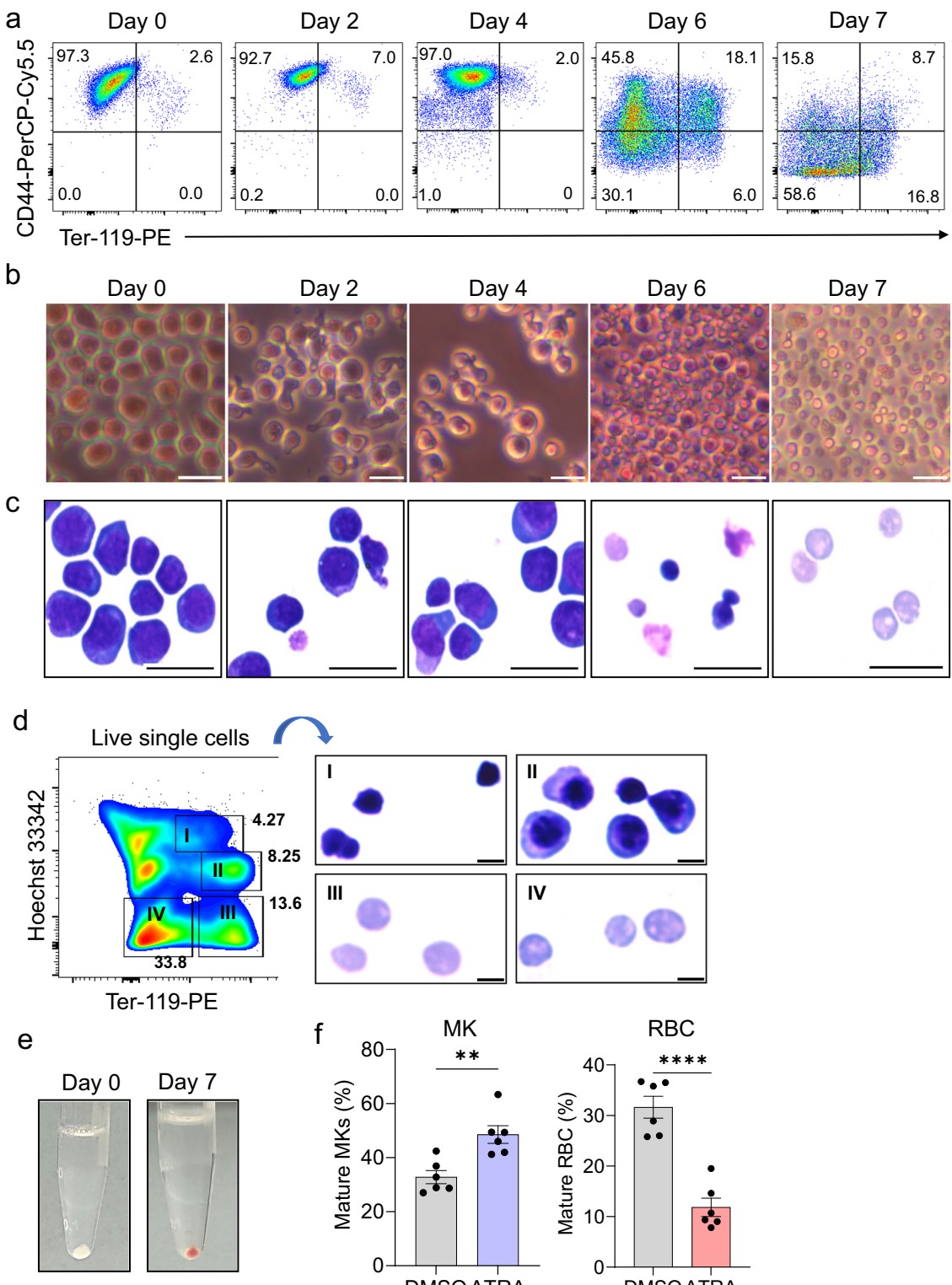

**Fig. 7 | In vitro cell differentiation of Hoxa7-TPO cells in the presence of EPO.**
**a** Flow cytometry analysis of in vitro differentiating Hoxa7-TPO cells in the presence of EPO. **b**, **c** Phase contrast (b)- and May-Grünwald-Giemsa (c)-based microscopy images of in vitro differentiating Hoxa7-TPO cells in the presence of EPO. Magnification 40x. Scale bars = 20 µm. **d** Enucleation analysis of Hoxa7-TPO cells that were differentiated for 7 days in the presence of EPO, followed by FACS and May-Grünwald-Giemsa-based microscopy. Gating strategy (left); Sorted cell fractions (right) are labeled with roman letters. I, extruded nuclei; II, nucleated cells; III, IV, Ter119high and Ter119low reticulocytes. Scale bars, 5 µm. **e** Cell pellets of Hoxa7-TPO cells left non-differentiated (left) and in vitro differentiated for 7 days in the presence of EPO (right). Red color indicates hemoglobin synthesis during in vitro differentiation. **f** Flow cytometry-based quantification of mature MK (CD41high CD42dhigh) and RBC (FSC-Alow SSC-Alow CD44low CD41-) that were derived from Hoxa7-TPO cells after 5 days in the presence of EPO (1 ng/ml) and TPO (1 ng/ml), treated either with DMSO or ATRA (50 nM). Data represent mean ± SEM from $n = 6$ independent experiments. Two-tailed unpaired t test, **$P = 0.0032$, ****$P = 0.000036$. Cohen's $d = 2.23$ for MK and 4.06 for RBC. Source data are provided as a Source Data file.

accompanied by a progressive reduction of cell size, as apparent by microscopy using phase contrast and May-Grünwald-Giemsa-staining (Fig. 7b, c). Flow cytometry sorting based on staining for Ter119 and the DNA dye Hoechst 33342, followed by cytospin/ May-Grünwald-Giemsa staining revealed late stage nucleated erythroblasts, enucleated cells (reticulocytes) and extruded nuclei (pyrenocytes) (Fig. 7d). As expected, cells differentiated under these conditions displayed a red cell pellet, reflecting hemoglobinization (Fig. 7e). We also tested the formation of burst forming units-erythroid (BFU-E) and colony forming units-erythroid (CFU-E) using the commercial Mega-Cult system (StemCell), but did not observe such colonies. We speculate that colony formation may depend on factors absent in our in vitro-culture system (provided by the in vivo BM environment), which will however require further studies. In any case, Hoxa7-TPO cells display RBC potential in vivo and in vitro.

### Alteration of cell cycle kinetics biases the differentiation of Hoxa7-TPO cells

Cell cycle kinetics can influence MEP differentiation toward RBC (cell cycle activation) or MK (cell cycle slowing)[14]. To investigate if Hoxa7-TPO cells recapitulated this MEP property, we established cell culture conditions suitable for simultaneous differentiation of both MK and RBC, and added all-trans retinoic acid (ATRA), which downregulates cyclins and cyclin-dependent kinases, reducing cell cycle progression[22]. Indeed, ATRA treatment increased the number of Hoxa7-TPO-derived MK and, simultaneously, reduced the number of RBC precursors (Fig. 7f). As noted, these data are also consistent with described ontogeny analysis, showing up- and down-regulated mitotic cell cycle-related genes in Hoxa7-TPO-derived RBC and MK, respectively (Supplementary Fig. 5). Thus, Hoxa7-TPO cells differentiate according to cell cycle cues in a similar manner as primary MEP[14].

### Hoxa7-TPO cells as tool for CRISPR/Cas9-mediated loss-of-function studies

An important future application for Hoxa7-TPO cells will be to analyze gene function by genetic modification. This includes generation of Hoxa7-TPO cells from BM of genetically modified mice or, alternatively, the genetic modification of established wild-type Hoxa7-TPO cells by gain- or loss-of-function approaches, including retroviral overexpression or CRISPR/Cas9-mediated deletion of genes of interest. To test this concept, we used CRISPR/ Cas9-mediated non-homologous end joining to disrupt three different genes with important functions in RBC or platelet development: (i) Nfe2, which encodes a transcription factor that is essential for normal MK and platelet production[20,23]; (ii), Gp1ba, which encodes the CD42b subunit of the platelet vWF receptor and is mutated in Bernard-Soulier syndrome, a macrothrombocytopenia with platelet dysfunction and bleeding diathesis[24–27]; and (iii) Klf1, which encodes a transcription factor that positively regulates differentiation of MEPs into erythroid precursors and their maturation into RBCs. Biallelic disruption of Klf1 causes defective erythropoiesis and increased platelet numbers[28,29]. We achieved efficient targeting of Nfe2, Gp1ba and Klf1 with Cas9 and specific gene-targeting small guide RNAs (sgRNAs)(Supplementary Fig. 10). Disruption of Nfe2 or Gp1ba led to strong reductions in platelet numbers without significant impact on RBC generation, consistent with the known functions of these genes (Fig. 8a). Reduced platelet numbers from GP1BA-deficient cells were accompanied by increased cell size, as apparent by IF analysis and flow cytometry, recapitulating the platelet phenotype of the Bernard-Soulier syndrome (Fig. 8b, c and Supplementary Fig. 11). Abnormally increased platelet size and reduced platelet production were particularly profound with Gp1ba sgRNA-2, which was associated with a higher targeting efficiency than Gp1ba sgRNA-1. Contrasting with the effects of Gp1ba or Nfe2 on platelet production, disruption of Klf1 in Hoxa7-TPO cells led to a strong reduction of RBC output in vivo and

significantly increased platelet numbers, similar to observations in Klf1-targeted mice (Fig. 8a)[28,30–33]. Together, these findings demonstrate that Hoxa7-TPO cells represent a facile model for examining human blood disorders and for analyzing efficiently the gene regulatory networks that control platelets and RBC formation from MEPs.

## Discussion

This work describes a simple method to generate hematopoietic progenitor cells lines with erythroid and MK potential. Similar to earlier methods that target WBC precursors, this approach is based on regulated Hox gene expression in combination with a defined growth factor, which block cell differentiation and promote cell proliferation, respectively[34]. An important observation from these studies was that the growth factor present during ex vivo expansion of Hox gene-expressing cells resulted in the expansion of specific progenitors from a mixed BM population[5]. For example, while GM-CSF in context with regulated Hox gene led to expansion of a macrophage precursor, FLT3L promoted expansion of a multipotent progenitor cell type with myeloid and lymphoid potential[34]. Consistent with the current study, the TPO receptor is expressed on MEPs, but not committed myeloid/ lymphoid progenitors, which likely explains how TPO selectively expands Hox-expressing MEP-like cells, while other progenitors and more differentiated cells are lost during cell propagation. Although there was some variation related to the output of RBC and platelets in vivo, all cell lines established and, importantly, all clones derived thereof, exhibited dual lineage potential, suggesting an overall homogenous nature of established cell lines. Homogeneity is also supported by transcriptional analyses, which further supports the close relationship of Hoxa7-TPO with MEP and a more distant relationships to all other progenitor cell types. While karyotype analysis indicated that Hoxa7-TPO cells are genetically stable for up to one year, functional analyses including cell cloning and sgRNA treatment, were restricted to ~8 weeks of continuous in vitro cell culture. As for any other cell line, it seems advisable to restrict cell culture periods as far as possible to avoid changes in key parameters, such as differentiation potential.

Whether the Hoxa7 gene is specifically required for MEP expansion or if other Hox genes such as Hoxb8, can be used to expand cells with similar lineage potential, remains unexplored. Likewise, whether the approach described here can be used to conditionally immortalize human MEP requires further study. In general, the factors governing Hox gene specificity remain largely unexplored. This is at least in part due to their low expression levels, their short residence time on chromatin, their cooperation with other proteins regulating specificity and, not least, the low number of endogenous, hematopoietic cells available for investigation[34]. Regarding the latter aspect, the Hoxa7-TPO cell line described here may be a useful tool to define Hoxa7-specific functions in MEPs.

Upon repression of Hoxa7 by estrogen withdrawal, Hoxa7-TPO cells differentiate via a series of defined precursors into mature RBC and platelets, which are largely indistinguishable from their primary counterparts with respect to morphology, gene transcription and function, e.g. platelet aggregation and thrombus formation. As such, this method facilitates the analysis of erythroid and MK cell differentiation and the functional investigation of the respective mature cell progeny. Various cellular systems to investigate MK and/or RBC development have been established. This includes MK cell lines (MEG-01, K562, 3T3-L1 preadipocytes), embryonic stem (ES)-cell/ induced pluripotent stem cell (iPS)-derived MKs, conditionally immortalized MK progenitors and ex vivo isolated, primary cells. These systems, however, have limitations related to partial or defective terminal cell differentiation (MK cell lines), complexity and duration of cell generation (ES/ iPS cells), restriction to the MK/ platelet lineage

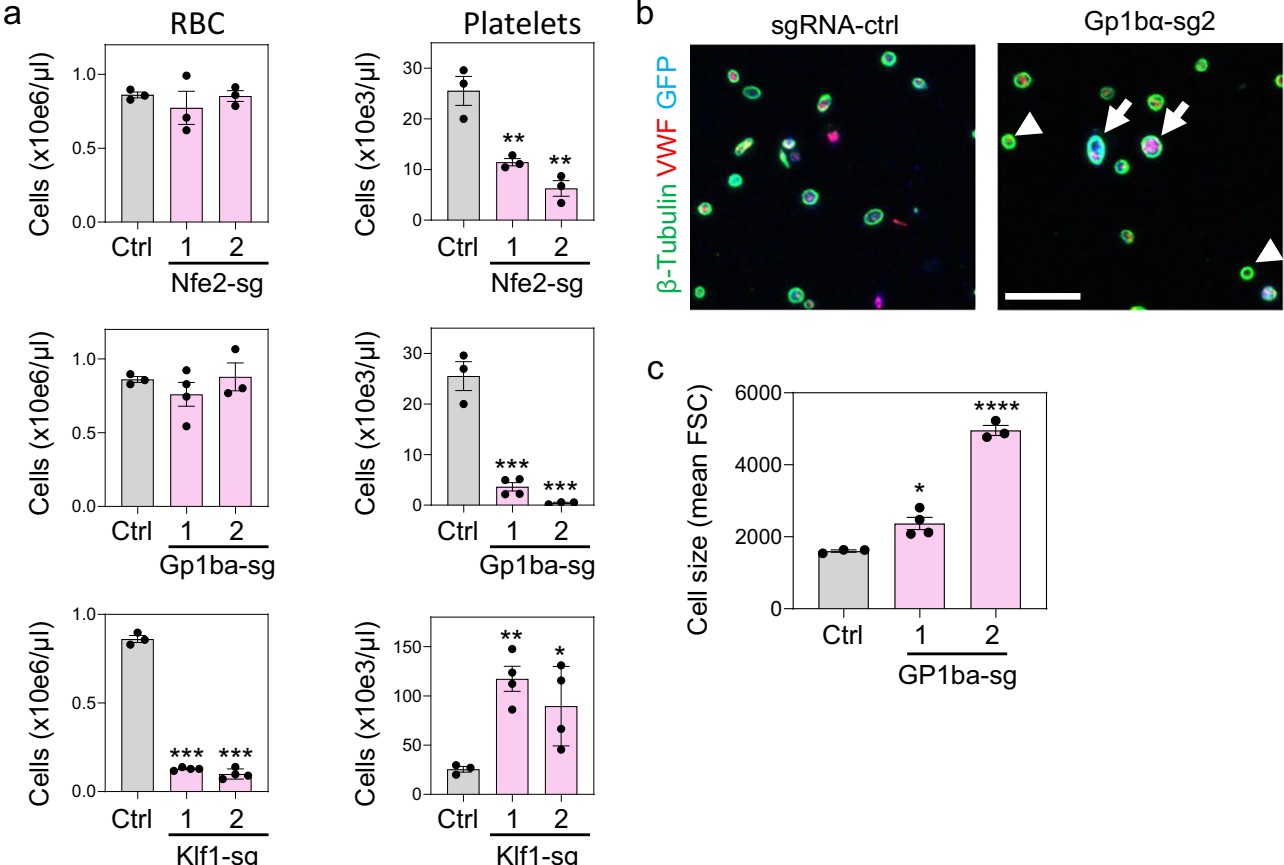

**Fig. 8 | CRISPR/ Cas9-mediated deletion of key differentiation factors. a** RBC (left) and platelet (right) numbers in the PB of mice that were adoptively transferred with GFP$^+$ Hoxa7-TPO cells 7 days after transfer. Hoxa7-TPO cells were transduced with expression vectors for Cas9 and sgRNAs against *Nfe2*, *Gp1ba* or *Klf1* as indicated. Cell numbers were determined by flow cytometry and CBC analysis. Data represent mean ± SD for n mice: Ctrl $n = 3$, Nfe2-sg1 $n = 3$, Nfe2-sg2 $n = 3$, Gp1ba-sg1 $n = 4$, Gp1ba-sg2 $n = 3$, Klf1-sg1 $n = 4$, Klf1-sg2 $n = 4$. *$P < 0.05$, **$P < 0.01$, ***$P < 0.001$; Two-tailed unpaired t test. Source data are provided as a Source Data file. **b** Immuno-fluorescence analysis of washed platelets derived from PF4-Cre-iDTR mice that were transferred with Hoxa7-TPO cells transduced with expression

vectors for Cas9 and control sgRNA (sgRNA-ctrl) or sgRNA targeting *Gp1ba* (Gp1ba-sg2). Platelets were seeded on chamber slides coated with poly-L-lysine and stained with antibodies against β-tubulin, VWF and GFP, followed by confocal microscopy. Arrows indicate GFP$^+$, Hoxa7-TPO-derived platelets, arrow heads indicate residual endogenous (GFP$^-$) platelets. Scale bars = 10 μm. **c** The cell size of platelets derived from the PB of mice that were adoptively transferred with Hoxa7-TPO cells transduced with Gp1ba-sgRNA (and control-sgRNA) was determined by flow cytometry. Data represent mean ± SD for n mice: Ctrl $n = 3$, Gp1ba-sg1 $n = 4$, Gp1ba-sg2 $n = 3$. *$P = 0.0136$, ****$P < 0.0001$. Two-tailed unpaired t test. Source data are provided as a Source Data file.

(conditional MK progenitors) or relatively low cell numbers available for manipulation and analysis (primary cells)[35–41]. Related to the erythroid lineage, similar systems have been established, including cell lines (MEDEP, HUDEP, HiDEP, EJ, BEL-A, BEL-C, JK-1), ES/ iPS-derived RBC and isolation of primary cells[42–47]. These systems, however, show similar limitations related to their terminal differentiation commitment (cell lines), complexity and time constraints (ES/ iPS cells) and relatively low cell numbers available (primary cells). Last, various leukemic cell lines with erythroid and megakaryocytic potential are available, such as K562, HEL and MEG-01, whose differentiation can be induced by compounds, such as PMA, hemin or butyric acid. However, these cell lines harbor various, partly characterized mutations, exhibit only limited terminal differentiation capacity and applications are limited to in vitro experiments[48–52]. To the best of our knowledge, Hoxa7-TPO cells represent the only cell system that (i) provides essentially unlimited numbers of bi-potent cells closely related to MEP, a cell type that is exceedingly rare and difficult to isolate and (ii) is amenable to facile genetic manipulation followed by in vitro and in vivo applications. The capacity of Hoxa7-TPO cells to faithfully recapitulate MEP's physiology is evidenced by our findings that their bi-potential differentiation is influenced by disruption of *Klf1* and interference with cell cycle progression, both of which favor

megakaryopoiesis. In summary, our findings indicate that this cell system has a wide range of possible applications to explore the biology of MEP, erythroid and MK lineages.

## Methods

### Reagents and plasmids
The following monoclonal antibodies were used in flow cytometry and immunofluorescence: TER-119-Pecy7 (TER-119), CD41-BV421 (MWReg30), CD34-PE (SA376A4), CD16/CD32-PE (93), CD11c-PerCP-Cy5.5 (N418), CD42d-APC (1C2); B220-PE (RA3-6B2), CD62P- PerCP-eFluor710 (Psel.KO2.3), CD115-APC (AFS98), CD11b-BV421 (M1/70), donkey anti-mouse IgG- Alexa Fluor™ 488, all donkey anti-rabbit IgG- Alexa Fluor™ 488 (all from Thermo Fisher Scientific); CD8a-APC (53-6.7), CD4-APC (RM4-5), C-KIT-APC (2B8), CD71-PE (C5) (all from Becton Dickinson). Sca-1-PE (E13-161.7), LY6G-Pecy7 (1A8), Ly6C-APC (HK1.4), CD150-APC (TC15-12F12.2), CD105-Pecy7 (MJ7/18), CD44-PerCP-Cy5.5 (IM7) and CD42d- Percp-cy5.5 (1C2) (all from Biolegend). Anti-GFP-Alexa Fluor™ 647 (Boster Bio). Anti-β1-tubulin (Sigma). Anti-VWF (Santa Cruz Biotech). Anti-CD41a-Alexa Fluor 647 (clone MWReg30, BD Biosciences).

The following reagents were used: β−estradiol (Sigma-Aldrich), recombinant mouse growth factors SCF, TPO, IL−3, IL−6, EPO

(Peprotech), Hirudin (Aniara Diagnostic), fibronectin (Sigma-Aldrich), Hoechst H33342 (Thermo Fisher Scientific), ATRA (Sigma-Aldrich), Alexa Fluor™ 488-Phalloidin conjugate (Thermo Fisher Scientific). A list of reagents used in this study is provided in Supplementary Data 3.

Mouse *Hoxa7* cDNA was cloned into MSCVneo (BD Biosciences) downstream of a triple hemagglutinin (HA) epitope tag. The hormone binding domain of the mutagenized estrogen receptor (ERHBD G400V) was cloned in frame between HA-tag and *Hoxa7*, generating the final MSCV–ERHBD–Hoxa7 vector. The correct sequence of the expression cassette was confirmed by DNA sequencing.

## Virus production

The plasmids MSCV–ERHBD–Hoxa7 and the ecotropic packaging vector pCL–Eco (Imgenex) were co-transfected into HEK293T cells using Lipofectamine 2000 (Invitrogen). 18 hours after transfection, the supernatant (SN) was replaced by fresh growth medium (DMEM (Invitrogen), supplemented with 10% (v/v) fetal bovine serum (FBS, Hyclone), 50 mM 2-mercaptoethanol, antibiotics (penicillin G (100 IU/ml) and streptomycin sulfate (100 IU/ml) and Pyruvate (1 mM)), followed by incubation for 24 hours and collection of virus- containing supernatant. Replication-deficient lentivirus was generated using Lipofectamine 2000 (Invitrogen)–based transient transfection of HEK293T cells using a four-plasmid system that was generously provided by I. Verma. Cas9-mediated deletion of genes in Hoxa7-TPO cells was done on the basis of cells transduced with the lentiviral CAS9 expression vector lentiCas9-Blast (Addgene, no. 52962)[53]. Single-guide RNAs (sgRNAs) were expressed by lentiviral delivery of specific sgRNA using the lentiviral vector lentiGuide-Puro (Addgene, no. 52963)[53]. Transduced cells were selected with puromycin (10 µg/ml) and used as polyclonal cell populations.

## Generation and cell culture of Hoxa7-TPO progenitor cell lines

BM cells were harvested by flushing femurs and tibias of 4–8 week old female mice with 10 ml RP10 (RPMI 1640 (Invitrogen), supplemented with 10 % (v/v) FBS, 50 mM 2–mercaptoethanol and antibiotics (penicillin G (100 IU/ml) and streptomycin sulfate (100 IU/ml)), pelleted by centrifugation, resuspended in 4 ml RP10, loaded on 3 ml Ficoll-Paque (GE Healthcare) and separated by centrifugation at 450 g for 30 min. The entire supernatant was collected (discarding only 500 µl including the cell pellet), diluted with 45 ml of PBS containing 1% FBS, pelleted at 800 g for 10 min, followed by resuspension in 10 ml RP10, centrifugation at 450 g for 5 min and resuspension at a concentration of $5 \times 10^5$ cells/ml in RP10 containing recombinant mouse IL–3 (10 ng/ml), IL–6 (20 ng/ml) and 100 ng/ml recombinant SCF (R&D Systems). After two days of cell culture, cells were collected and resuspended in progenitor outgrowth medium (POM), i.e., X-VIVO10 (Lonza) containing 10 % (FBS, Hyclone), 50 mM 2-mercaptoethanol, antibiotics (penicillin G (100 IU/ml) and streptomycin sulfate (100 IU/ml)), supplemented with 1 µM β–estradiol and murine TPO (20 ng/ml). $3 \times 10^5$ cells were dispensed in 1 ml per well in a 12–well plate and infected with the MSCV–ERHBD–Hoxa7 vector by spin inoculation at 1500 g for 60 min in the presence of Lipofectamine (0.1 %, Invitrogen). After spinoculation, cells were diluted by adding 1.5 ml POM for 24 h, followed by replacement of 2 ml of the cell culture medium with fresh POM. During the following cell culture period, cells were transferred to new wells containing fresh medium every 3–4 days. Once the cell populations were stably expanding, cells were kept at concentrations between $1 \times 10^5$ and $1.5 \times 10^6$ cells/ml.

## Mouse strains and adoptive transfer of Hoxa7-TPO cells

All mouse studies were carried out in accordance with protocols approved by the Institutional Animal Care and Use Committees at the St. Jude Children's Research Hospital, the University of Utah and the Children's Hospital of Philadelphia. Transgenic mice with pancellular

EGFP expression under control of an H2Kb enhancer/promoter were generously provided by Derek Persons[15]. PF4-Cre-iDTR transgenic mice were generated as previously described by crossing the PF4-Cre strain (Jackson Laboratories, stock #008535) with the ROSA26iDTR strain (Jackson Laboratories, stock #007900)[54–56]. $Mpl^{-/-}$ mice were obtained from Dr. Wei Tong at the Children's Hospital of Philadelphia[19]. For in vivo reconstitution experiments, $5 \times 10^6$ GFP+ Hoxa7-TPO cells, together with $2 \times 10^5$ BM cells from C57BL/6 J mice were transferred into lethally irradiated C57BL/6 J recipient female mice via tail vain or retro-orbital injection. Irradiation doses were 950 rad (cesium-137) for transfer experiments into wt mice, $2 \times 500$ rad (Radsource RS2000) for experiments using PF4-Cre-iDTR mice and 1000 rad (XRAD320, Precision) for experiments using $Mpl^{-/-}$ mice. At the different time points after the transfer, PB was collected by retro-orbital bleeding, or mice were sacrificed and organs were harvested. Complete blood counts (CBC) were measured using the Forcyte Hematology System (Oxford Science). Single cell suspensions were analyzed by flow cytometry.

## Adoptive transfer of Hoxa7-TPO cells for immunofluorescence analysis

For analysis of endogenous and Hoxa7-TPO-derived platelets, PF4-Cre-iDTR mice (6–8 weeks old) were injected intraperitoneally with one dose of diphtheria toxin (DT, 400 ng), followed by two additional doses of DT (200 ng) administered every other day to deplete endogenous MKs and platelets. One day after the last DT treatment, mice were lethally irradiated ($2 \times 500$ rad at an interval of 3 hours, Radsource RS2000), followed by intravenous injection of Hoxa7-TPO cells (2 x 10e7)(day 0) along with unfractionated BM cells (2 x 10e5) from PF4-Cre-iDTR mice that had been treated with DT using the same regimen. Additional Hoxa7-TPO cells (2 x 10e7) were transferred on day 1. On day 6, whole blood was collected via cardiac puncture. Blood from various experimental groups was pooled and washed platelets were isolated in Tyrodes buffer.

For analysis of platelets derived from sgRNA-treated Hoxa7-TPO cells, PF4-Cre-iDTR mice (8–12 weeks old) were injected intraperitoneally with a single dose of DT (400 ng). Three days after DT treatment, mice were lethally irradiated ($2 \times 500$ rad at an 3 hour interval, Radsource RS2000) and transplanted with 4 x 10e7 control or Gp1ba-deleted Hoxa7-TPO cells, along with BM cells (2 x 10e5) from PF4-Cre-iDTR mice that had undergone the same DT treatment (day 0). On day 5, whole blood was collected via cardiac puncture. A fraction of PB from each mouse was used for flow cytometry analysis. The remaining blood from each control and experimental group was pooled and used for immunofluorescence analysis.

## Immunofluorescence analysis of platelets

Chamber slides were coated with 200 µL of poly-L-lysine (0.1 mg/mL) or fibrinogen (100 µg/ml) in a 37 °C humidified chamber for 1 hr. Slides were rinsed 3x with PBS and blocked with BSA (0.1 %) in PBS for 30 min. Washed platelets were prepared in modified Tyrodes buffer at a density of 2 x10e7 cells/mL left to rest for 30 minutes. 1.5x10e6 total platelets per slide were spread on coated cover-glass for 45 minutes at 37 °C under humidified conditions. Unbound platelets were washed away (3x gentle PBS washes) and adhered cells were fixed using paraformaldehyde (4 %) in PBS for 10 minutes at room temperature, followed by blocking with FBS (2 %) in PBS for 1 h. Immunostaining was performed using primary antibodies specific to β1-tubulin and vWF at 1:100 dilutions overnight at 4 °C under humidified conditions. Chamber slides were washed 3x with PBS and incubated with Alexa Fluor™ 488 donkey Anti-Mouse IgG, Alexa Fluor™ 488 donkey Anti-Rabbit secondary antibodies at a 1:1000 dilution, and an anti-GFP Alexa Fluor™ 647 at a 1:100 dilution for 45 min at room temperature in the dark. Platelets spread on fibrinogen were stained using Alexa Fluor™ 488-Phalloidin conjugate at a 1:50 dilution and anti-GFP Alexa Fluor™

647 at a 1:100 dilution. Chamber slides were washed 3x with PBS, and mounted with coverslips using Prolong Glass before curing overnight in the dark. Images were obtained using a Leica TCS SP5 II Confocal microscope and a 60x oil objective.

### Intravital cremaster arteriole laser injury studies

These studies were conducted using a confocal intravital imaging system as previously described[18]. A 100 µl bolus of Alexa Fluor 647-labeled anti-CD41 F(ab')2 (0.3 µg/g mouse, clone MWReg30, BD Bioscience) in PBS was infused into the mice via a jugular vein cannula to visualize platelets. Arterioles (20-40 µm diameter) were injured using an SRS NL100 pulsed nitrogen dye laser (440 nm) aimed at the endothelial lining. Brightfield and timelapse images were captured on the injuries made in each mouse and analyzed using Slidebook 6.0 software (Intelligent Imaging Innovations) to determine the area and fluorescent intensity of platelets.

### Activation of platelets in whole blood

Platelet activation was determined as described previously[57]. In brief, 500 µl blood was drawn from the vena cava inferior using a 1 ml syringe containing 50 µl sodium citrate (3.2%).40 µl of whole blood were seeded into half-area 96-well microtiter plates (Greiner Bio-One) coated with hydrogenated gelatin (0.75% w/v, Sigma). For stimulation, collagen (Chronolog) was added at a final concentration of 2 µg/ml and incubated for 5 min at 37 °C on a plate shaker at 1200 rpm. 5 µl of treated blood were transferred into 45 µl acid citrate dextrose solution (5 mM glucose, 6.8 mM trisodium citrate, 3.8 mM citric acid) and 50 µl PBS/0.5% FBS containing CD42d-APC were added and incubated for 20 min, followed by addition CountBright Absolute counting beads (Thermo Fisher Scientific) and flow cytometry analysis. The percent loss of single platelets was calculated using the formula: [unstimulated platelet number minus collagen-stimulated platelet number]/ unstimulated cd42d positive platelet number x 100.

### Activation of Platelets in platelet rich plasma (PRP)

1 ml mouse blood was drawn from the vena cava inferior using a 1 ml syringe containing 100 µl Sodium Citrate (3.2 %). 2 ml of wash buffer (13.4 mM NaCl, 0.3 mM KCL, 0.03 mM NaH2PO4, 0.5 mM HEPES, 0.5 mM Glucose, 0.2 mM MgCl2, 11 mM NaHCO3, 2 mg/ml BSA, 1 mM EGTA, pH 6.8) was added, and blood was centrifuged at 190 g for 3 min. The top layer representing PRP was transferred to a new 15 ml falcon tube and prostaglandin E1 (PGE1) was added to a final concentration of 1 µM to prevent premature platelet activation. Cells were collected by centrifugation (1450 g, 8 min) and resuspended in 1 ml of final resuspension buffer (13.4 mM NaCl, 0.3 mM KCL, 0.03 mM NaH2PO4, 0.5 mM HEPES, 0.5 mM Glucose, 0.2 mM MgCl2, 11 mM NaHCO3, 2 mg/ml BSA, pH 7.2). 2 µl of washed platelets were added to 98 µl of final resuspension buffer containing antibodies against CD62P, CaCl2 (1 mM) and Thrombin (2 U/ml (Sigma-Aldrich)) and incubated for 20 min at 37 °C. Thereafter, samples were diluted with 200 µl of resuspension buffer and analyzed immediately by flow cytometry.

### Hemoglobin fractionation

RBC sorted by Fluorescence-activated Cell Sorting (FACS) were lysed in Hemolysate reagent (Helena Laboratories) and collected by centrifugation (1000 g,10 min). Analysis of globin chains in the supernatant was performed by ion exchange (IE) and reverse phase (RP)-HPLC (Prominence UFLC, Shimadzu). Proteins eluting from the column were quantified by light absorbance at 220, 280, and 415 nm with a diode array detector.

### In vitro MK differentiation and proplatelet generation

Hoxa7-TPO cells were washed twice with PBS containing FBS (2 %) and cultured for different periods of time in POM supplemented with TPO (20 ng/ml) to generate MK. MK were evaluated by phase contrast

microscopy, cytospin and May-Grünwald-Giemsa-staining and flow cytometry as described in figure legends. MK ploidy was determined by flow cytometry based on Hoechst H33342 staining. For proplatelet production, TPO (50 ng/ml) and hirudin (100 anti-thrombin units) were added to Hoxa7-TPO or BM cultures throughout cell differentiation (4 days), and mature MKs were isolated using a BSA gradient and seeded at 1x10e5 cells per well in a 24-well plate, either left uncoated (for phase contrast microscopy) or coated with fibrinogen (100 µg/ml)(for proplatelet quantification and DIC microscopy)[58]. Proplatelet formation was assessed 12 h later using phase-contrast or DIC microscopy.

### In vitro erythroid cell differentiation and enucleation analysis

Hoxa7-TPO cells were washed twice with PBS containing 2% FBS and seeded into 12-well plates coated with fibronectin (2.5 µg per square centimeter) at a density of 2x10e5 cells per well in 2 ml POM supplemented with EPO (1 ng/ml). On day 6 after plating, the medium was replaced with 2 ml POM containing 20% FBS without EPO for 24 hours. Cells were analyzed by flow cytometry and May-Grünwald-Giemsa staining. For enucleation assessment on day 7, cells were stained with antibodies against Ter-119 and Hoechst 33342, followed by flow cytometry analysis. Various fractions, including extruded nuclei, as well as nucleated and enucleated cell populations, were FACS-sorted, stained with May-Grünwald-Giemsa, and analyzed by microscopy.

### Dual MK/ Erythroid differentiation assay and cell cycle inhibition

To evaluate the effect of cell cycle kinetics on the lineage bias of Hoxa7-TPO cells, an optimized dual megakaryocyte (MK) and erythroid (E) differentiation assay was used. Hoxa7-TPO cells were washed twice with PBS containing FBS (2 %) and seeded into uncoated 12-well plates (2 x 10e5 cells per well) in POM (2 ml) containing TPO (1 ng/ml) and EPO (1 ng/ml). In some conditions ATRA (50 nM) or its solvent DMSO was added. Five days later, cells were analyzed by flow cytometry to quantify MK (CD41$^+$ CD42d$^+$) and erythroid cells (FSC-A$^{low}$ SSC-A$^{low}$ CD44$^{low}$ CD41$^-$).

### Computational analysis of RNA-seq data from Hoxa7-TPO cell lines

Public murine microarray data (GSE77098) was imported into Partek Genomics Suite 6.6 and rma normalized. A Kruskal Wallis test was applied to each probeset across selected categories. Those data that passed the Bonferroni threshold at 5 percent were retained. Data were then imported into STATA/MP 15.1 and filtered to retain the highest expressing probe-set per gene symbol. Each gene was then assigned to the signature category that had the highest median expression. Next, the data were matched by gene symbol to RNAseq log 2 FPKM data, where values for genes within a signature were averaged for each signature and each RNAseq sample. These sample signatures were then z-score normalized across samples, creating a −2 to 2 scale. The resultant z-scores were subsequently imported and visualized in R version 3.5.1 using the pheatmap package. RNAseq data of non-differentiated and differentiated Hoxa7-TPO cells are available from NCBI GEO (GSE244976).

### Computational analysis of lineage-defining factors and over-representation analysis

Raw RNA-seq reads were first processed with TrimGalore (v0.6.10) to remove adapters and low quality reads. Filtered reads were then mapped to the mouse genome (GRCm39.112) with the STAR aligner (v2.7.11a), using the "--quantMode GeneCounts" option to generate gene-level read counts. Raw counts were then processed using the R package DESeq2 (v1.42.1) for differential expression analysis comparing undifferentiated Hoxa7-TPO cells to differentiated ones (MK and RBC, separately). Genes with a Benjamini-Hochberg corrected p-

value < 0.05 were considered significantly differentially expressed. These genes were then interpolated to experimentally defined bone-marrow MK and RBC markers reported in the CellMarker 2.0 database[59]. Differentially expressed genes were also used for an over-representation analysis of Gene Ontology "Biological Process" terms (Fisher's exact test), with upregulated and downregulated genes analyzed separately. Gene Ontology terms with a Benjamini-Hochberg corrected p-value < 0.05 were considered significantly enriched. The code used for the over-representation analysis is available at Github (https://github.com/mmarchetti90/project_setup_assistant/blob/main/code_base/term_enrichment/ora/enrichment_analysis_1.py)

### Quantitative PCR (Q-PCR)

Total RNA was extracted using the RNeasy Plus Universal Mini Kit (Qiagen) and reverse-transcribed into cDNA using the iScript™ cDNA Synthesis Kit (Bio-Rad). Q-PCR was performed using the ViiA™ 7 Real-Time PCR System (Applied Biosystems) with SYBR™ Green dye (Applied Biosystems). The primer sequences used were as follows: *Vwf*: Forward: TCATCGCTCCAGCCACATTCCATA, Reverse: AGCCACGC TCACAGTGGTTATACA; *Pf4*: Forward: TTCTGGGCCTGTTGTTTC TG, Reverse: GATCTCCATCGCTTTCTTCG; *Gapdh*: Forward: AGG TTGTCTCCTGCGACTTCA, Reverse: CCAGGAAATGAGCTTGACAAAG. Gene expression levels were quantified using the standard curve method and normalized to *Gapdh* expression.

### Analysis of sgRNA targeting efficiency

DNA was extracted from cells using the PureLink Genomic DNA Mini Kit (Life Technologies). A first-round PCR amplification was conducted using a 2x Phusion High-Fidelity PCR Master Mix using gene-specific primers. A second PCR amplification reaction was performed with primers containing sample-specific adapters (Illumina). PCR products were sequenced on a MiSeq platform (Illumina) with 2 ×150-bp paired-end reads. Sequence alignment and mutation detection were performed using CRISPResso2 software. The gRNA sequences and first-round PCR primers are listed in Supplementary Table 1.

### Statistics and reproducibility

Statistical analyses were performed using data from at least three independent biological and/or technical replicates. Data distribution was assessed using the Shapiro-Wilk test. Group differences were evaluated using two-tailed unpaired t-tests or one-way analysis of variance (ANOVA) followed by Tukey's test for multiple comparisons. For select pairwise comparisons, Cohen's *d* was calculated post hoc to estimate effect size and aid interpretation. No statistical method was used to predetermine sample size. No data were excluded from the analyses. The experiments were not randomized. The Investigators were not blinded to allocation during experiments and outcome assessment. Statistical significance was defined as $P < 0.05$, with significance levels indicated as follows: *$P < 0.05$; **$P < 0.01$; ***$P < 0.001$; ****$P < 0.0001$.

### Reporting summary

Further information on research design is available in the Nature Portfolio Reporting Summary linked to this article.

## Data availability

Relevant data supporting the findings of this study are available upon request from the corresponding author, respectively have been deposited at NCBI GEO under accession code GSE244976. Source data are provided with this paper.

## Code availability

The code used for this study is available at GitHub [https://github.com/mmarchetti90/project_setup_assistant] and has been archived on Zenodo at https://doi.org/10.5281/zenodo.15626059 (release v1.0.0)[60].

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

## Acknowledgements

This work was supported by the American Lebanese Syrian Associated Charities (ALSAC), NIH grant R35HL150698 to M.P., NIH grants RO1 AI145877 and RO1 AI170942 to H.H.

## Author contributions

R.W. and F.S. performed experiments, analyzed data and compiled part of the Material and Methods section. Z.M. and V.R. performed experiments. M.M. and D.F. performed bioinformatics analyses. K.A.Q. and A.C.P. planned, conducted and analyzed ex vivo platelet morphology experiments. C.O.F. and H.S.A. performed the arteriole injury model and analyzed data. M.P. provided guidance for in vivo platelet experiments. Y.C. analyzed sgRNA targeting efficiency. P.X. and M.J.W. conducted

hemoglobin analysis. H.H. conceived and supervised the project, performed experiments, analyzed data, and wrote the manuscript. All authors edited the manuscript.

## Competing interests

The authors declare no competing interests.
