## [Transparent Peer Review file · Nature Communications]

Murine hematopoietic progenitor cell lines with erythroid and megakaryocyte potential

Corresponding Author: Dr Hans Haecker

Version 0:

Reviewer comments:

Reviewer #1

(Remarks to the Author)
General comments

1. The same group led by Dr. Hacker (correspondence author) had previously demonstrated the same concept of hematopoietic progenitor cell lines by Hoxb8 dependent cell proliferation system (reference 4: Nat Methods, 10(8), 795, 2013). This paper in 2013 depicted that Hoxb8 with Flt3L lost self-renewal capacity and potential to differentiated into megakaryocytes (MK) and erythrocytes (Ery) but retain the potential to myeloid lineage including macrophages, granulocytes, dendric cells, and lymphoid lineage by B lymphocytes and T lymphocytes. Although the author stated “we focused on HoxA7 due to its known, physiological function in MEP development (reference 7: Blood, 103, 3192, 2004), there is no apparent scientific link between Hoxb8 effectors and Hoxa7 effectors in the hierarchy of hematopoietic progenitor development from hematopoietic stem cell population. I think this paper lacks such most important information regarding how Hox family based regulation system in hematopoietic system. The exact contributions and mechanisms of Hoxa7 and TPO in directing the differentiation towards RBCs and platelets are not well-defined.
2. The manuscript lacks in vivo functional study. Does the time to stop bleeding differ between mice implanted with and without Hoxa7-TPO?
3. Information regarding the recovery and viability of the Hoxa7-TPO cells after long-term cryopreservation is absent, which is vital to ascertain their utility as a research tool.

Specific comments

1. The study lacks adequate control groups, such as in vivo endogenous cells, which is crucial for validating the experimental results.
2. To enhance the transparency of the RNA-seq analysis, it is recommended that the authors provide complete gene ontology information for both up- and down-regulated gene lists. Additionally, the expression profiles of various transcriptional factors and pathways involved in MK and RBC differentiation should be presented to offer a comprehensive view.
3. What gene sets are driven by Hoxa7 overexpression? What genes and downstream effectors from Hoxa7 contribute to the promotion of MEP self-renewal?
4. The authors demonstrated a long-term culture of 1 year. However, there is no information of any change in the growth rate, various surface markers or differentiation ability into MK/Ery lineages even after long-term culture?
5. Hoxa7-TPO cells cultured in TPO in the presence of estrogen are negative for both Ter119 and CD41. When these cells were cultured with TPO in the absence of estrogen, multinucleation and formation of proplatelet-like structures were observed?
6. When differentiation is induced in the absence of both estrogen and TPO, do they differentiate into Ery? What happens when EPO is added?
7. The study lacks a detailed functional analysis, especially in vitro. No strong functional evidence was shown supporting RBC maturation from Hoxa7-TPO cells in vitro.
8. Figure 1E: the scale bars mentioned are not visible.
9. Figure 3B: the methodology behind creating the gene expression heatmap is unclear. Specifying whether FPKM values were used in this analysis would be helpful.

10. Figure 4A: the gating strategy defining platelets and RBCs is unclear. And there is no information on RBCs that they are denucleated, and it seems in this manuscript that the Ter119(+) RBCs include nucleated cells. In contrast, Ter119(-) RBCs appear to contain other haemocytic cells.
11. Figures 4B and DE: the differences between the two figures are not apparent. The improvement with clearer demarcation is required.
12. Figure 7: in the text depicting Figure 7, there is replicate.
13. While the paper notes changes in platelet size due to Gp1b-alpha editing, it does not provide visual evidence to substantiate this observation.

Reviewer #2

(Remarks to the Author)

The manuscript by Wu and coworkers presents a new method to immortalize MEP precursors induced by a murine stem cell virus (MSCV)-based retroviral vector containing *Hoxa7*, whose expression is under the control of a modified estrogen-binding domain. They show that the cells can survive for more than a year in the presence of estrogen+TPO. They also characterize in vitro the phenotype of MKs obtained after estrogen withdrawal. In this population, cell size increases as well as ploidy, hallmarks of MKs. These cells were able to extend some protrusions resembling classical in vitro proplatelets and expressed a number of MK markers. In vivo, injection of *Hoxa*-TPO cells into lethally irradiated mice resulted in the formation of platelets and red blood cells characterized by expression of markers. Finally, the authors used Crispr/Cas9 in these MEPs to delete a number of genes known to affect platelet or RBC production. After adoptive transfer, the mice showed the characteristic defects expected from the introduced sgRNA against the different genes.

The author's finding is very interesting as it describes a way to easily obtain immortalized MEP capable of differentiating in vitro and also in vivo into MKs and RBCs. This will be an invaluable tool for researchers in the field. The manuscript is well written and the data are of high quality, they mainly presenting the characterization of the immortalized cells and the cells derived after differentiation.

The authors should however be more precise and add a number of important details for cell characterization.

Figure 2. What is the kinetics of MK generation after estrogen withdrawal? Please provide the flow cytometry dot plot, the gating strategy, and the percentage of "MK" in the culture.

What is the proportion of MK-expanding proplatelets? Compare to native progenitors.

Figure 4. Is the size of day7 GFP+ platelets increased compared to untransfected mice? And compared to mice transferred with isogenic bone marrow cells?

Figure 4D and 4E: Is there a correlation between clones that give low/high platelet counts and clones with low/high RBC counts?

Figure 5. The platelet characterization provided is minimal. It is unclear if the data are limited to GFP+ platelets or whole PRP including platelets from the co-injected BM helper cells. Please clarify, and if whole PRP, perform the experiment focusing only on GFP+ platelets.

It would be of interest to provide data on the morphology of these *Hoxa7*-TPO derived platelets to visualize the microtubule coil as a hallmark of non-activated platelets and the presence of granules, best by transmission electron microscopy or at least after immunolabeling of tubulin and anti-VWF or PF4.

Are these in vivo generated GFP+ platelets able to incorporate a thrombus in an in vivo thrombosis model?

Figure 7. Similar to Figure 5, characterization is lacking, such as cell images, TEM and/or IF showing the main feature of platelets and/or RBC differentiated from progenitors defective for the targeted genes.

Minor points:

Introduction: correct the sentence: "Given the key effector functions of RBCs and platelets, i.e. oxygen transport and blood coagulation, respectively". In fact, the role of platelets is primarily in "primary hemostasis" and not in "coagulation" per se, although they may also contribute, but to a lesser extent.

Reviewer #3

(Remarks to the Author)

In this paper, the authors report the generation of a murine cell line, *HoxA7*-TPO, capable of both megakaryocyte and erythroid differentiation. *HoxA7*-TPO cells were generated by the expression of an estrogen-driven *HoxA7* gene and exogenous thrombopoietin receptor. Removal of estrogen with continued TPO in the media caused these cells in vitro to become polyploid and express a megakaryocyte gene signature. Importantly, in vivo delivery of GFP+ *HoxA7*-TPO cells to lethally irradiated mice resulted in a transient production of both platelets and red blood cells, consistent with their identity as megakaryocyte-erythroid progenitors (MEP). The circulating GFP+ platelets were capable of placing P-selectin on their surface following agonist activation. A wave of erythroid lineage differentiation was identified in the marrow and the resulting GFP+ red cells circulated for several weeks. Finally, *HoxA7*-TPO cells were edited via Crispr to disrupt either *NFE2L3*, *GP1bB*, or *Klf1* and the cells transferred into irradiated mouse recipients. Disruption of platelet numbers in the first two, and decreased red cells with increased platelets in the third, were found, as expected of the known lineage-specific function of these genes.

Strengths of this paper include the logical lay-out of the experimental data, the novel use of *HoxA7* to immortalize cells in the megakaryocyte/erythroid lineage pathway, the unique MEP stage of hematopoietic immortalization, and the ability to genetically modify these cells to transiently create potential disease models following transfer of these cells into myeloablated mice. In the Discussion the authors nicely compare *HoxA7*-TPO cells with various immortalized and leukemic-derived megakaryocyte and erythroid cell lines and highlight the unique MEP stage of *HoxA7*-TPO cells. They conclude the

Discussion stating that HoxA7-TPO cells have “a wide range of possible applications to explore the biology of erythroid and megakaryocyte lineages”.

Several questions and concerns are raised, in part focusing on the MEP nature of HoxA7-TPO cells:

First, consistent with their MEP identity, are HoxA7-TPO cells capable of forming megakaryocyte and erythroid colonies when cultured in vitro, e.g., in collagen?

Second and importantly, are HoxA7-TPO cells capable of erythroid differentiation in vitro or only in vivo?

Third, building on studies of MEP, does alteration of cell cycle kinetics bias the differentiation of HoxA7-TPO cells toward a megakaryocyte or erythroid fate?

Fourth, since mpl and EPO-receptor are members of the group 1 cytokine receptor sub-family, do HoxA7-TPO cells respond to exogenous EPO and does this exposure predispose these cells toward an erythroid differentiation pathway?

Fifth, are HoxA7-TPO cells capable of platelet-like particle formation when differentiated in vitro?

Minor issues:

1. The final paragraph of the results section is duplicated. It is presumed that the second one was meant to be included.
2. Figure 4A, the identity of the GFP+ and GFP- Ter119-negative cells remains unclear. The authors speculate that these represent a “less mature RBC”. How immature must a RBC be to be Ter119-? What is the evidence that they are, in fact, red blood cells? Could these cells belong to some other hematopoietic lineage? While Ter119- red cells have been reported following in vitro differentiation, are their reports of Ter119- red blood cells present in the bloodstream in vivo?
3. Regarding the in vivo generated red blood cells, does HPLC measure “hemoglobin content”?
4. The authors mention culturing these cells for a year. A Supplemental graph with growth kinetics over time would be helpful.

Version 1:

Reviewer comments:

Reviewer #1

(Remarks to the Author)

Although the authors have previously employed this system in various cell types (such as DCs and HPCs), those studies were primarily methodological and similarly did not explore mechanistic underpinnings. Unfortunately, this revised submission follows the same pattern, lacking significant new insights beyond the technical aspects.

Moreover, the manuscript still suffers from issues that suggest a lack of thorough revision. Spelling and formatting errors remain (e.g., on page 5, the phrase “...which may contribute to differences in lineage output” and inconsistent fonts in figure legends such as Fig. 2g,h). Additionally, the FACS plot in Fig. 2g (day 5) contains out-of-scale data, which hampers proper interpretation.

There are also several inconsistencies and ambiguities that were not resolved:

Fig. 1a presents a contradiction between the legend and the main text regarding the use of control cells.

The potential inhibitory effect of estrogen on control cells was not addressed, and a comparative growth curve in the absence of estrogen remains missing.

The abstract contains misleading phrasing, implying the discovery of a novel Hoxa7 isoform rather than a synthetic regulatory system.

In Fig. 4b, the meaning of the different symbols is unclear and should be clarified—do they represent distinct experiments or cell populations?

The qPCR methodology is insufficiently described; essential details such as the internal control gene are omitted.

Fig. 5a lacks an antibody isotype control, which is necessary given the unexpectedly high signal in untreated cells.

Most concerning is the conflicting information between the main text and the Methods section regarding the use of thrombin versus prostaglandin E1—compounds with opposing biological effects in platelet activation assays.

Finally, the authors suggest that their system may be applied in rare cell populations for loss-of-function studies. While this is a potentially valuable application, it remains entirely speculative in the absence of any supporting data, such as experiments using patient-derived samples.

Reviewer #2

(Remarks to the Author)

The reviewers have carefully answered all my points and performed some rigorous and nice additional works. My only left suggestion is that they indicate in the title and abstract that they are working with mouse cells

Reviewer #3

(Remarks to the Author)

The authors have responded in a thorough and thoughtful manner to the critiques and suggestions of the reviewers. Importantly, they have performed several new experiments incorporating significant new data into the manuscript, particularly new in vivo data revealing the functionality of HoxA7-TPO-derived platelets in clot formation and in vitro erythroid differentiation studies of HoxA7 cells. Additionally, new experimental evidence is provided supporting the concept that cell cycle regulation in MEPs is an important factor in downstream lineage choice. These new data significantly strengthen the conclusions of this paper and bring a new cell line into the repertoire of investigators who study the megakaryocyte and erythroid lineages.

Rebuttal letter

Response to Reviews

We thank the referees for their thorough review and constructive criticism. We addressed all points raised, in large part by additional experiments, which we feel have indeed significantly improved our manuscript and substantiate the conclusions Hoxa7-TPO cells will serve as a particularly versatile tool to investigate cell differentiation and biology of MEPs and erythroid and megakaryocyte lineages.

Changes in the revised manuscript are marked in red.
Please find below our detailed point-by-point rebuttal.

Reviewer #1 (Remarks to the Author):

General comments

1. The same group led by Dr. Hacker (correspondence author) had previously demonstrated the same concept of hematopoietic progenitor cell lines by Hoxb8 dependent cell proliferation system (reference 4: Nat Methods, 10(8), 795, 2013). This paper in 2013 depicted that Hoxb8 with Flt3L lost self-renewal capacity and potential to differentiated into megakaryocytes (MK) and erythrocytes (Ery) but retain the potential to myeloid lineage including macrophages, granulocytes, dendritic cells, and lymphoid lineage by B lymphocytes and T lymphocytes. Although the author stated “we focused on HoxA7 due to its known, physiological function in MEP development (reference 7: Blood, 103, 3192, 2004), there is no apparent scientific link between Hoxb8 effectors and Hoxa7 effectors in the hierarchy of hematopoietic progenitor development from hematopoietic stem cell population. I think this paper lacks such most important information regarding how Hox family based regulation system in hematopoietic system. The exact contributions and mechanisms of Hoxa7 and TPO in directing the differentiation towards RBCs and platelets are not well-defined.

Response: We certainly appreciate this question, but believe that understanding the specificity of different Hox genes in hematopoiesis is really outside the scope of this paper (please see detailed explanation below). However, given that this is an interesting topic, we have modified the manuscript and discuss this important point in various parts of the manuscript.

Detailed explanation:

The differential contribution of *Hoxb8* and *Hoxa7* to hematopoiesis, respectively self-renewal of specific genes and progenitor cell populations, is certainly a very interesting question. It is, however, also one of the most challenging ones that has, to the best of our knowledge, not been resolved for any of the Hox genes. This scarcity of information is at least in part related to the technical difficulties to investigate Hox genes by ChIP assays, which is (despite their remarkable biological impact) reflected by only a handful of published examples and complete absence of HOX ChIP-seq data in genomic consortia, such as ENCODE. As detailed by Luo et al (PMID: 30866492), technical challenges may relate to the very low expression of Hox genes (in fact it is challenging to detect even retrovirally expressed 3HA-ERHBD-Hoxa7 by Western blotting) and

their short residence time on chromatin, possibly requiring modifications of standard ChIP assays, e.g. by including different protein-protein-crosslinkers (in addition to formaldehyde)(PMID: 27974206). Still, even the rudimentary data available from HOX ChIP assays suggest (i) that HOX proteins bind primarily distal regulatory elements within intergenic and intronic regions (>20 Kb from a transcription start site)(reviewed in PMID: 30604842) and (ii) that specificity is likely conferred by additionally recruited proteins that interact specifically with certain HOX proteins. These technical difficulties are further increased by the extremely low cell numbers of progenitors available for analysis. As such, delineation of Hoxa7 vs. Hoxb8 biology would really constitute an entirely novel project entailing likely the development of new technology related to ChIP, mass spectrometry (for identification of interacting proteins) and, not least, extensive follow-up studies to confirm the significance of differentially bound gene regions. As noted above, we believe it is fair to say that this goes well beyond the scope of this paper, whose focus is methods development to generate MEP-like cells to allow for investigation of cell differentiation and Mk/ erythroid biology. The data presented (including a large set of new data generated for this revision) clearly demonstrate (i) the close relationship between Hoxa7-TPO cells/ cell progeny and their primary counterparts (MEP, MK, RBC) and (ii) the overall versatility of the system to investigate MEP/MK/RBC differentiation and biology.

2. The manuscript lacks in vivo functional study. Does the time to stop bleeding differ between mice implanted with and without Hoxa7-TPO?

Response: Analysis of the bleeding time is technically difficult due to the fact (i) that Hoxa7-TPO-derived platelets are obtained at significantly lower than physiological levels (~10% in a typical experiment) and (ii) that there are still a certain number of endogenous platelets found in the peripheral blood that may contribute to hemostasis, even under various conditions we established for this revision to reduce endogenous platelets as far as possible, including irradiation, genetic MPL-deletion and diphtheria toxin (DT)-mediated deletion in PF4-DTR mice (in combination with irradiation)(shown in the revised manuscript).

To overcome these issues and directly analyze the *in vivo* function of Hoxa7-TPO-derived platelets, we now conducted experiments to measure the contribution of Hoxa7-derived platelets to thrombus formation using a cremaster arteriole injury model. These data show that Hoxa7-TPO-derived platelets indeed contribute to thrombus formation, as expected (Fig. 5c-g of revised paper), which suggest that Hoxa7-TPO-derived platelets are as effective or more effective in clotting than endogenous platelets.

3. Information regarding the recovery and viability of the Hoxa7-TPO cells after long-term cryopreservation is absent, which is vital to ascertain their utility as a research tool.

Response: We investigated this important aspect and found that freezing and storing Hoxa7-TPO cells in liquid nitrogen did not reduce viability and cell performance as assessed by recovery and adoptive transfer of Hoxa7-TPO cells that had been cryopreserved for 4 years in liquid nitrogen (Supplementary Fig. 8).

Specific comments

1. The study lacks adequate control groups, such as in vivo endogenous cells, which is crucial for validating the experimental results.

Response:

We added various additional direct control groups, including endogenous RBC, BM-derived megakaryocytes and platelets, which further illustrates the high degree of similarity between Hoxa7-TPO-derived and endogenous cells.

Figure 2: establish BM-Mk cultures and perform phase contrast and cytopins. May need to do this in direct comparison to Hoxa7-TP-derived Mk to make sure that microscope-images (including May-Grünwald staining) are comparable.

Response: These control images are added to the revised paper. As suspected by the reviewer, new cytopins for both, Hoxa7-TPO- and BM-derived MK populations were required to make them directly comparable.

2. To enhance the transparency of the RNA-seq analysis, it is recommended that the authors provide complete gene ontology information for both up- and down-regulated gene lists. Additionally, the expression profiles of various transcriptional factors and pathways involved in MK and RBC differentiation should be presented to offer a comprehensive view.

Response: We performed gene ontology analyses of differentially expressed genes (MK vs. non-differentiated cells and RBC vs. non-differentiated cells), as well as a gene expression analysis of lineage-defining transcription factors for the various cell populations, as requested (Supplementary Fig. I-Fig. 4 and 5).

3. What gene sets are driven by Hoxa7 overexpression? What genes and downstream effectors from Hoxa7 contribute to the promotion of MEP self-renewal?

Response: Please see response to general comment 1. While we agree with the notion that this will be interesting, we feel that it is really a project on its own with likely considerable challenges due to the complexity of Hox gene analysis in general.

4. The authors demonstrated a long-term culture of 1 year. However, there is no information of any change in the growth rate, various surface markers or differentiation ability into MK/Ery lineages even after long-term culture?

Response: This extended period of cell culture was really just intended to test genomic stability under somewhat extreme conditions. By no means would we recommend to extend cell culture beyond time periods required for specific experiments, most of which should not exceed 8 weeks of cell culture. This time frame is what we tested in various experimental settings, including single cell sorting and sgRNA treatment, confirming proper cell growth and differentiation. We clarify this point explicitly in the revised discussion. We still added the growth curve of these long-term cultures, which are presented in Supplementary Fig. 1.

5. Hoxa7-TPO cells cultured in TPO in the presence of estrogen are negative for both Ter119 and CD41. When these cells were cultured with TPO in the absence of estrogen, multinucleation and formation of proplatelet-like structures were observed?

Response: Yes, this is correct. Multi-nucleation and pro-platelet formation are shown in Fig. 2b,c. Pro-platelet-like structures are shown at higher magnification in Fig. 2d. We include now also a DIC image of pro-platelets (which are particularly nice and plastic, Fig. 2e), as well as

experiments exploring pro-platelet formation quantitatively in Hoxa7-TPO- vs. BM-derived MKs for comparison (Fig. 2f).

6. *When differentiation is induced in the absence of both estrogen and TPO, do they differentiate into Ery? What happens when EPO is added?*

Response: In the absence of growth factor the cells die quickly as they are strictly growth factor-dependent, comparable to primary progenitor cells (see Fig. 1d). However, when adding EPO, the cells survive and indeed differentiate in a time-dependent fashion into RBC as shown in the revised manuscript in detail by flow cytometry and microscopy (Fig. 7). Interestingly, a substantial fraction of cells extrudes their nuclei, resulting in mature, enucleated RBC, even *in vitro* (Fig. 7d), accompanied by hemoglobin production (Fig. 7e).

7. *The study lacks a detailed functional analysis, especially in vitro. No strong functional evidence was shown supporting RBC maturation from Hoxa7-TPO cells in vitro.*

Response: This evidence has been added; please see response to comment 6.

8. *Figure 1E: the scale bars mentioned are not visible.*

Response: This was an oversight, which has been corrected (this figure does not contain a scale bar).

9. *Figure 3B: the methodology behind creating the gene expression heatmap is unclear. Specifying whether FPKM values were used in this analysis would be helpful.*

Response: This is now properly described in both legends and methods section.

10. *Figure 4A: the gating strategy defining platelets and RBCs is unclear. And there is no information on RBCs that they are denucleated, and it seems in this manuscript that the Ter119(+) RBCs include nucleated cells. In contrast, Ter119(-) RBCs appear to contain other haemocytic cells.*

Response:

The gating strategy is now highlighted in more detail in the modified Fig. 4a. The de-nucleated character of Hoxa7-TPO-derived Ter119(+) RBC is highlighted in the new Fig. 4e, which shows completely de-nucleated reticulocytes. We have now also investigated in more detail the nature of Hoxa7-TPO-derived Ter119(-) RBC, which are found in relatively high percentages at early time points (~40% of Hoxa7-TPO-derived cells at day 7 after transfer), but are progressively reduced during later time points (~18% at day 42 after transfer). As shown in the new Fig. 4e, both Ter119(+) and Ter119(-) RBC populations consist homogeneously in enucleated cells (endogenous RBC are shown for comparison). Along with mentioned up-regulation of Ter119 on Hoxa7-TPO-derived RBC during time, it appears that the Ter119(-) RBC indeed represent mature RBC, as hypothesized, although Ter119 is not (yet?) expressed. Whether this reflects particularly slow maturation of Hoxa7-TPO-derived RBC or, alternatively, reflects cell maturation under emergency hematopoiesis, remains to be investigated. In any case, Hoxa7-TPO-derived Ter119(-) RBC represent morphologically mature RBC. Based on these FACS analyses, we have no indication that other, non-RBC cells are contained in the RBC-gate. Please

also note that no other cell type is found at cell numbers comparable to RBC, which makes this possibility extremely unlikely.

11. Figures 4B and DE: the differences between the two figures are not apparent. The improvement with clearer demarcation is required.

Response: The differences between the figures, i.e. platelets/ RBC derived from adoptively transferred polyclonal populations (Fig. 4B) and platelets/ RBC derived from adoptively transferred cell clones, is now highlighted in the title ('derived from clones') in the revised Fig. 4g/h.

12. Figure 7: in the text depicting Figure 7, there is replicate.

Response: We were unfortunately not able to find the replicate. Maybe this was a PDF file artifact?

13. While the paper notes changes in platelet size due to Gp1b-alpha editing, it does not provide visual evidence to substantiate this observation.

Response: We performed an additional set of data to analyze the morphology of Hoxa7-TPO-derived platelets (including those generated by GP1Ba-deficient ones) by immunofluorescence, in direct comparison to primary, respectively control platelets. These data show the classic morphology (and actin-dependent spreading characteristic) of platelets (Fig. 4c,d), and also illustrates the substantially increased size of GP1Ba-deficient platelets (Fig. 8b)(along with flow cytometry data, Fig. 8c).

Reviewer #2 (Remarks to the Author):

The manuscript by Wu and coworkers presents a new method to immortalize MEP precursors induced by a murine stem cell virus (MSCV)-based retroviral vector containing Hoxa7, whose expression is under the control of a modified estrogen-binding domain. They show that the cells can survive for more than a year in the presence of estrogen+TPO. They also characterize in vitro the phenotype of MKs obtained after estrogen withdrawal. In this population, cell size increases as well as ploidy, hallmarks of MKs. These cells were able to extend some protrusions resembling classical in vitro proplatelets and expressed a number of MK markers. In vivo, injection of Hoxa-TPO cells into lethally irradiated mice resulted in the formation of platelets and red blood cells characterized by expression of markers. Finally, the authors used Crispr/Cas9 in these MEPs to delete a number of genes known to affect platelet or RBC production. After adoptive transfer, the mice showed the characteristic defects expected from the introduced sgRNA against the different genes.

The author's finding is very interesting as it describes a way to easily obtain immortalized MEP capable of differentiating in vitro and also in vivo into MKs and RBCs. This will be an invaluable tool for researchers in the field. The manuscript is well written and the data are of high quality, they mainly presenting the characterization of the immortalized cells and the cells derived after differentiation.

The authors should however be more precise and add a number of important details for cell

characterization.

Figure 2. What is the kinetics of MK generation after estrogen withdrawal? Please provide the flow cytometry dot plot, the gating strategy, and the percentage of "MK" in the culture. What is the proportion of MK-expanding proplatelets? Compare to native progenitors.

Response: In the revised manuscript, we provide the gating strategy (Supplementary Fig. 3) and a detailed kinetics analysis of Hoxa7-TPO cells during cell differentiation using flow cytometry and microscopy (Fig. 2g-j), collectively illustrating the remarkable homogeneity of the resulting MK population. We also performed quantitative analyses of MK-producing pro-platelets, which show comparable results for Hoxa7-TPO- and BM-derived MK (Fig. 2f).

Figure 4. Is the size of day7 GFP+ platelets increased compared to untransfected mice? And compared to mice transferred with isogenic bone marrow cells?

Response: We directly compared Hoxa7-TPO-derived platelets with endogenous, BM-derived platelets and platelets from un-treated mice and didn't find significant differences (Supplementary Fig. 6)

Figure 4D and 4E: Is there a correlation between clones that give low/high platelet counts and clones with low/high RBC counts?

Response:

We analyzed these data and found indeed a decent correlation with a Pearson correlation index of $r = 0.46$. While these experiments were primarily intended to investigate if cell clones retain both MK and RBC potential (which most clones do), it seems that the overall potential varies to some extent. However, we note that the cloning process itself (in our experience) can add quite some stress to obtained cell populations, which may contribute to differences in lineage potential. The correlation analysis is now included as Supplementary Fig. 7 in the revised manuscript.

Figure 5. The platelet characterization provided is minimal. It is unclear if the data are limited to GFP+ platelets or whole PRP including platelets from the co-injected BM helper cells. Please clarify, and if whole PRP, perform the experiment focusing only on GFP+ platelets.

Response: This experiment is unfortunately technically extremely difficult, if not impossible. We tried various conditions to completely eliminate endogenous cells, however, under none of the conditions tested, including *Mpl*^{-/-} mice and PF4-DTR mice (even in combination with irradiation), endogenous platelets were completely deleted. As such, we were not able to perform the experiment exactly as requested. In this context it should be noted that 'endogenous' platelets correspond with very high likelihood to residual cells due to the 3-5-day half-life in mice, rather than originating from transferred BM cells. Only 2×10^5 BM 'helper' cells are transferred in our experiments to avoid BM failure after lethal irradiation. As such, it is extremely unlikely that this unfractionated population of cells contributes measurably to platelet production 5 days after transfer.

Importantly, to assess the function of Hoxa7-TPO-derived platelets more directly, we now conducted *in vivo* thrombus formation assays in *Mpl*^{-/-} mice in an arteriole laser injury model. These data (Fig. 5c-g) clearly demonstrate the contribution of Hoxa7-TPO-derived platelets to

thrombus formation and even to enhance incorporation of endogenous, MPL-deficient platelets into thrombi. These data, along with another set of new data characterizing the morphology of Hoxa7-TPO-derived platelets *ex vivo* (please see below), strongly suggest that the cells are functional.

It would be of interest to provide data on the morphology of these Hoxa7-TPO derived platelets to visualize the microtubule coil as a hallmark of non-activated platelets and the presence of granules, best by transmission electron microscopy or at least after immunolabeling of tubulin and anti-VWF or PF4.

Response: We performed immunofluorescence analyses as this allows us to differentiate Hoxa7-TPO-derived cells from (residual) endogenous cells using antibodies to GFP. In these experiments we used antibodies to tubulin, VWF (and GFP) as suggested, and also included fibrinogen-coated slides to analyze spreading of Hoxa7-TPO-derived platelets (in comparison to endogenous cells). These data, which are shown in Fig. 4c-d of the revised manuscript, show a comparable morphology of Hoxa7-TPO-derived and endogenous platelets, including the characteristic tubulin-ring and fibrinogen-induced spreading.

In these experiments we employed PF4-DTR mice, whose MKs (and thus platelets) can be deleted by diphtheria toxin, increasing the percentage of Hoxa7-TPO-derived cells (albeit not completely eliminating them, as mentioned).

Figure 7. Similar to Figure 5, characterization is lacking, such as cell images, TEM and/or IF showing the main feature of platelets and/or RBC differentiated from progenitors defective for the targeted genes.

Response: We performed the same set of experiments as described in the previous point (pooling six mice to obtain sufficient numbers of platelets from Gp1ba-deficient Hoxa7-TPO cells, which are very rare). The new data demonstrate the enlarged phenotype of Gp1ba-deficient platelets corresponding to platelets observed in the Bernard-Soulier syndrome, as expected (Fig. 8b).

Minor points:

Introduction: correct the sentence: "Given the key effector functions of RBCs and platelets, i.e. oxygen transport and blood coagulation, respectively". In fact, the role of platelets is primarily in "primary hemostasis" and not in "coagulation" per se, although they may also contribute, but to a lesser extent.

Response: This has been corrected.

Reviewer #3 (Remarks to the Author):

In this paper, the authors report the generation of a murine cell line, HoxA7-TPO, capable of both megakaryocyte and erythroid differentiation. HoxA7-TPO cells were generated by the expression of an estrogen-driven HoxA7 gene and exogenous thrombopoietin receptor. Removal of estrogen with continued TPO in the media caused these cells in vitro to become polyploid and express a megakaryocyte gene signature. Importantly, in vivo delivery of GFP+ HoxA7-TPO cells to lethally irradiated mice resulted in a transient production of both platelets and red blood cells, consistent with their identity as megakaryocyte-erythroid progenitors (MEP). The

circulating GFP+ platelets were capable of placing P-selectin on their surface following agonist activation. A wave of erythroid lineage differentiation was identified in the marrow and the resulting GFP+ red cells circulated for several weeks. Finally, HxA7-TPO cells were edited via Crispr to disrupt either NFE2, GP1bB, or Klf1 and the cells transferred into irradiated mouse recipients. Disruption of platelet numbers in the first two, and decreased red cells with increased platelets in the third, were found, as expected of the known lineage-specific function of these genes.

Strengths of this paper include the logical lay-out of the experimental data, the novel use of HoxA7 to immortalize cells in the megakaryocyte/erythroid lineage pathway, the unique MEP stage of hematopoietic immortalization, and the ability to genetically modify these cells to transiently create potential disease models following transfer of these cells into myeloablated mice. In the Discussion the authors nicely compare HoxA7-TPO cells with various immortalized and leukemic-derived megakaryocyte and erythroid cell lines and highlight the unique MEP stage of HoxA7-TPO cells. They conclude the Discussion stating that HoxA7-TPO cells have “a wide range of possible applications to explore the biology of erythroid and megakaryocyte lineages”.

Several questions and concerns are raised, in part focusing on the MEP nature of HoxA7-TPO cells:

First, consistent with their MEP identity, are HoxA7-TPO cells capable of forming megakaryocyte and erythroid colonies when cultured in vitro, e.g., in collagen?

Response: We tested the performance of Hoxa7-TPO cells in colony forming assays using the commercial MegaCult system (StemCell) and found that the cells form typical, cholinesterase-positive MK-CFU when cultured in MegaCult-C medium in the presence of TPO, IL-3 and IL-6 (StemCell, Cat. #04960)(please see image below). However, under conditions typically used to generate BFU-E/ CFU-E (StemCell M3334, insulin, transferrin, EPO) or preferentially BFU-E (M3436, insulin, transferrin, EPO and unspecified cytokines), we did not find significant formation erythroid colonies. Of note, as shown in detail in the revised manuscript, when Hoxa7-TPO cells are cultured in bulk in medium containing EPO, the cells differentiate faithfully into mature Ter119-positive, hemoglobin-containing RBC, even with a significant fraction ejecting their nuclei. As such, the cells clearly have MK and RBC potential, which they realize both *in vivo* and *in vitro*. The question why the cells don't form CFU-B/ CFU-E comparable to BM-derived MEP remains unclear. While obviously not related to lineage potential, it may be related to other factors -present in the BM environment, but not our in vitro system- allowing primary cells to form colonies. Given these uncertainties, we decided to leave these data out of the paper, but describe our observation in the results section.

Figure for reviewers only: Hoxa7-TPO cells form megakaryocyte colonies in semisolid medium.

A, B Hoxa7-TPO cells (A) and BM cells (B) were cultured in MegaCult-C medium (StemCell) for 8 days and colony formation was assessed by microscopy following cholinesterase staining.

Second and importantly, are HoxA7-TPO cells capable of erythroid differentiation in vitro or only in vivo?

Response:

The cells differentiate indeed nicely into Ter119+ RBC (including enucleated cells) *in vitro* when cultured in simple medium containing EPO (at an optimized concentration of 1 ng/ml). These data are described in detail in the revised manuscript (Fig. 7).

Third, building on studies of MEP, does alteration of cell cycle kinetics bias the differentiation of HoxA7-TPO cells toward a megakaryocyte or erythroid fate?

Response:

This is a very interesting point regarding an MEP-distinctive function. To address this question carefully, we established culture conditions that were conducive to development of both MK and RBC in the same sample (using optimized concentrations of TPO and EPO). Using these conditions and ATRA, we found indeed increased MK generation upon lowering cell cycle speed, accompanied by reduced RBC generation. These new data are shown in Fig. 7f and described in both text and discussion sections of the revised manuscript.

Fourth, since mpl and EPO-receptor are members of the group 1 cytokine receptor sub-family, do HoxA7-TPO cells respond to exogenous EPO and does this exposure predispose these cells toward an erythroid differentiation pathway?

Response: Yes, the cells are EPO responsive and differentiate into RBC upon EPO treatment as detailed in the revised manuscript (please see also response to second point)(Fig. 7).

Fifth, are HoxA7-TPO cells capable of platelet-like particle formation when differentiated in vitro?

Response: Yes, Hoxa7-TPO cells produce pro-platelets comparable to BM-derived MK as shown in the revised manuscript microscopically and quantitatively (Fig. 2d-f).

Minor issues:

1. *The final paragraph of the results section is duplicated. It is presumed that the second one was meant to be included.*

Response: Thank you - this duplication has been removed.

2. *Figure 4A, the identity of the GFP+ and GFP- Ter119-negative cells remains unclear. The authors speculate that these represent a “less mature RBC”. How immature must a RBC be to be Ter119-? What is the evidence that they are, in fact, red blood cells? Could these cells belong to some other hematopoietic lineage? While Ter119- red cells have been reported following in vitro differentiation, are their reports of Ter119- red blood cells present in the bloodstream in vivo?*

Response: To address the nature of these Ter119-negative cells, which reside in the RBC scatter gate, we sorted the cells, followed by cyto-spin/ microscopy analysis. As shown in the new Fig. 4e, both Ter119(+) and Ter119(-) RBC consist in homogenous populations of enucleated, classic RBC (endogenous, BM-derived RBC are shown for comparison). Along with the observation that Ter119(-) RBC are found at higher numbers early (~40% of Hoxa7-TPO-derived cells at day 7 after transfer), but are progressively reduced during later time points (~18% at day 42 after transfer), it appears that Ter119 expression is somewhat delayed on a fraction of these Hoxa7-TPO-derived RBC. Whether this reflects a particularly slow maturation of Hoxa7-TPO-derived RBC or, alternatively, reflects cell maturation under the particular conditions of emergency hematopoiesis in an irradiated BM environment, remains unclear. We did not find literature that would specifically address this issue, but have also not found such cells in ‘regular’ BM-based transplantation assays. However, as noted, the conditions are quite unique and we have not tested if transplantation of MEP that would mimic cell differentiation of Hoxa7-TPO cells most closely, could recapitulate this observation. Given that in vitro-generated RBC show notoriously reduced Ter119 expression, it seems possible that a proper BM environment is required to allow for swift Ter119 upregulation (which appears to reflect a rather complex co-expression of NXPE2 and GYPA (reported at the 2023 Ash meeting).

(<https://ashpublications.org/blood/article/142/Supplement%201/3817/503837/NXPE2-Is-the-Target-of-Ter-119-When-Complexed-with>)).

In any case, Hoxa7-TPO-derived Ter119(-) RBC represent morphologically mature RBC. We have no indication that other, non-RBC cells are contained in this cell population. Please also note that no other cell type is found at cell numbers comparable to RBC, which makes this possibility very unlikely.

We discuss these aspects appropriately in the revised manuscript.

3. *Regarding the in vivo generated red blood cells, does HPLC measure “hemoglobin content”?*

Response: We clarified this statement in the main text section.

4. The authors mention culturing these cells for a year. A Supplemental graph with growth kinetics over time would be helpful.

Response: We added comparative growth curves of such ‘old’ cell population and a regular, younger one to the supplementary data section (Supplementary Fig. 1a).

Of note, this extended period of cell culture was really just intended to test genomic stability under somewhat extreme conditions. By no means would we recommend to extend cell culture beyond time periods required for specific experiments, most of which should not exceed 8 weeks of cell culture. This time frame is what we tested in various experimental settings, including single cell sorting and sgRNA treatment, confirming proper cell growth and differentiation. We clarify this point explicitly in the revised discussion.

We thank the reviewers and editors again for their careful reviews and critiques of our manuscript.

Reviewer #1 (Remarks to the Author)

Although the authors have previously employed this system in various cell types (such as DCs and HPCs), those studies were primarily methodological and similarly did not explore mechanistic underpinnings. Unfortunately, this revised submission follows the same pattern, lacking significant new insights beyond the technical aspects.

Response: We thank the editors for overruling this ‘general concern’.

Moreover, the manuscript still suffers from issues that suggest a lack of thorough revision. Spelling and formatting errors remain (e.g., on page 5, the phrase "...which may contribute to differences in lineage output" and inconsistent fonts in figure legends such as Fig. 2g,h). Additionally, the FACS plot in Fig. 2g (day 5) contains out-of-scale data, which hampers proper interpretation.

Response:

‘Lineage’ has been corrected to ‘lineage’ and the capital font of panels g and h was changed to lowercase.

We respectfully disagree that the ‘out-of-scale data’ hamper proper interpretation of Fig. 2g. While it is correct that few cells are slightly above the top of the scale (due to the particularly high CD41 expression levels of large, differentiated MK), this really doesn’t change in any way data interpretation, i.e. that the cells up-regulate homogenously high levels of CD41 during cell differentiation. The small percentage of cells ‘out-of-scale’ is illustrated in the following histogram analysis (for reviewer only). This analysis demonstrates (i) that indeed only 6.6% of the cells are slightly out of scale and (ii) that these cells obviously express slightly higher (not lower) levels of CD41, as apparent from the curve of the histogram. Given that the data interpretation is not affected in any way (and changing the scale is technically not possible, unless the entire experiment would be repeated), we suggest to leave the figure as is.

Histogram analysis of day 5 of Fig. 2g illustrating the small percentage of cells (6.6%) expressing high levels of CD41 exceeding the flow cytometry scale.

There are also several inconsistencies and ambiguities that were not resolved:

Fig. 1a presents a contradiction between the legend and the main text regarding the use of control cells. The potential inhibitory effect of estrogen on control cells was not addressed, and a comparative growth curve in the absence of estrogen remains missing.

Response: We clarified this point in the results and legend sections: Control cells are cells transduced with an empty control vector that were cultured under the same conditions (including estrogen) as cells transduced with the estrogen-responsive Hoxa7 construct. As such, proliferation of Hoxa7-transduced

cells is unequivocally driven by Hoxa7. This point is further supported by various other data based on removal of estrogen, where Hoxa7-TPO cells homogenously differentiate into Mk or RBC, in vitro and in vivo.

We have now added a comparative growth curve of established Hoxa7-TPO cells that were cultured with TPO, either in the presence or absence of estrogen, resulting in continuous cells growth (in the presence of estrogen due to Hoxa7 activation) or cell differentiation into megakaryocytes (as detailed in Fig. 2) and cessation of cell growth (in the absence of estrogen), as expected. This growth curve is now added as Supplementary Fig. 3a to the revised manuscript.

The abstract contains misleading phrasing, implying the discovery of a novel Hoxa7 isoform rather than a synthetic regulatory system.

Response: We clarified this point by adding ‘virally transduced’ to the following sentence: Here we show that a virally transduced, regulated form of Hoxa7 can be used to conditionally immortalize and expand MEP-like cells (Hoxa7-TPO) that undergo erythro-megakaryocytic differentiation upon Hoxa7 inactivation.

In Fig. 4b, the meaning of the different symbols is unclear and should be clarified—do they represent distinct experiments or cell populations?

Response: The symbols indeed represent independently established Hoxa7-TPO cell populations. This point has been clarified in main text and figure legend.

The qPCR methodology is insufficiently described; essential details such as the internal control gene are omitted.

Response: We added a paragraph to the Methods section, which provides the details requested.

Fig. 5a lacks an antibody isotype control, which is necessary given the unexpectedly high signal in untreated cells.

Response: This experiment was actually done with appropriate isotype controls. They were not included in the original submission as there was no indication of any non-specific binding. The flow cytometry analysis based on these isotype control antibodies is now included as Supplementary Fig. 8d as requested.

Most concerning is the conflicting information between the main text and the Methods section regarding the use of thrombin versus prostaglandin E1—compounds with opposing biological effects in platelet activation assays.

Response: Prostaglandin E1 is used in the way as is custom in functional platelet assays, i.e. to prevent premature platelet activation during sample preparation. The assay itself measures thrombin-mediated platelet activation as correctly described in the manuscript. We modified the Methods section to clarify this point.

Finally, the authors suggest that their system may be applied in rare cell populations for loss-of-function studies. While this is a potentially valuable application, it remains entirely speculative in the absence of any supporting data, such as experiments using patient-derived samples.

Response: This is likely a misunderstanding. We specifically say that Hoxa7-TPO cells represent a ‘system that (i) provides essentially unlimited numbers of bi-potent cells closely related to **MEP, a cell type that is exceedingly rare and difficult to isolate** and (ii) is **amenable to facile genetic manipulation followed by in vitro and in vivo applications.**’ ‘Rare’ clearly refers to MEP, and our data based on sgRNA-mediated deletion of three different genes, i.e. Nfe2, Gp1ba and Klf1, and the detailed investigation of

their differentiation properties *in vivo* (Fig. 8) and experiments conducted *in vitro* (e.g. Fig. 7f) clearly support our interpretation that these cells can be used for various applications. Patient-derived samples cannot be used as this is a mouse-based system (clarified in title and abstract of the revised manuscript as requested by reviewer 2).

Reviewer #2 (Remarks to the Author)

The reviewers have carefully answered all my points and performed some rigorous and nice additional works. My only left suggestion is that they indicate in the title and abstract that they are working with mouse cells

Response: This point has been addressed in title and abstract as requested.

Reviewer #3 (Remarks to the Author)

The authors have responded in a thorough and thoughtful manner to the critiques and suggestions of the reviewers. Importantly, they have performed several new experiments incorporating significant new data into the manuscript, particularly new *in vivo* data revealing the functionality of HoxA7-TPO-derived platelets in clot formation and *in vitro* erythroid differentiation studies of HoxA7 cells. Additionally, new experimental evidence is provided supporting the concept that cell cycle regulation in MEPs is an important factor in downstream lineage choice. These new data significantly strengthen the conclusions of this paper and bring a new cell line into the repertoire of investigators who study the megakaryocyte and erythroid lineages.

Response: We thank the reviewer for the kind evaluation of our work.